

# Surface and tropospheric ozone over East Asia and Southeast Asia from observations: distributions, trends, and variability

Ke Li[1,*,#], Rong Tan[1,#], Wenhao Qiao[1,#], Taegyung Lee[2], Yufen Wang[1], Danyuting Zhang[1], Minglong Tang[1], Wenqing Zhao[1], Yixuan Gu[1], Shaojia Fan[3], Jinqiang Zhang[4], Xiaopu Lyu[5], Likun Xue[6], Jianming Xu[7,8], Zhiqiang Ma[9,10], Mohd Talib Latif[11], Teerachai Amnuaylojaroen[12], Junsu Gil[13], Mee-Hye Lee[13], Juseon Bak[14], Joowan Kim[15], Hong Liao[1], Yugo Kanaya[16], Xiao Lu[3], Tatsuya Nagashima[17], Ja-Ho Koo[2,*]

[1]Joint International Research Laboratory of Climate and Environment Change, Jiangsu Key Laboratory of Atmospheric Environment Monitoring and Pollution Control, Collaborative Innovation Center of Atmospheric Environment and Equipment Technology, School of Environmental Science and Engineering, Nanjing University of Information Science and Technology, Nanjing 210044, China

[2]Department of Atmospheric Sciences, Yonsei University, Seoul 03722, South Korea

[3]School of Atmospheric Sciences, Sun Yat-sen University, Zhuhai, Guangdong, China

[4]Key Laboratory of Middle Atmosphere and Global Environment Observation, Institute of Atmospheric Physics, Chinese Academy of Sciences, Beijing 100029, China

[5]Department of Geography, Faculty of Social Sciences, Hong Kong Baptist University, Hong Kong, China

[6]Environment Research Institute, Shandong University, Qingdao, China

[7]Shanghai Typhoon Institute, Shanghai Meteorological Service, Shanghai 200030, China

[8]Shanghai Key Laboratory of Meteorology and Health, Shanghai Meteorological Service, Shanghai 200030, China

[9]Institute of Urban Meteorology, China Meteorological Administration, Beijing 100089, China

[10]Beijing Shangdianzi Regional Atmosphere Watch Station, Beijing 101507, China

[11]Department of Earth Sciences and Environment, Faculty of Science and Technology, Universiti Kebangsaan Malaysia, Bangi, Selangor, Malaysia

[12]Atmospheric Pollution and Climate Change Research Units, School of Energy and Environment, University of Phayao, Phayao 56000, Thailand

[13]Department of Earth and Environment Sciences, Korea University, Seoul 02841, South Korea

[14]Institute of Environmental Studies, Pusan National University, Busan 46241, Republic of Korea

[15]Department of Atmospheric Sciences, Kongju National University, Kongju 32588, South Korea

[16]Japan Agency for Marine-Earth Science and Technology, Yokohama, Japan

[17]National Institute for Environmental Studies, Tsukuba 305-8506, Japan

[#]These authors contributed equally

*Correspondence to: Ke Li (keli@nuist.edu.cn) and Ja-Ho Koo (zach45@yonsei.ac.kr)



**Abstract.** High level of ozone throughout the troposphere is an emerging concern over East Asia and Southeast Asia. Here we analyzed available surface ozone measurements in the past two decades (2005-2021) over eight countries, and ten ozonesonde and aircraft measurements within this region. At surface, seasonal mean ozone over 2017-2021 varies from 30 ppb in Southeast Asia to 75 ppb in summer in North China. The metric of seasonal 95th percentile ozone can identify the multiple hotspots of ozone pollution of over 85 ppb in Southeast Asia. The new WHO peak season ozone standard indicates that both East Asia and Southeast Asia face a widespread risk of long-term exposure. The surface ozone increase in South Korea and Southeast Asia from 2005 was leveling off or even decreased in the past decade, while ozone increase in 2000s over China has amplified after 2013. Surface ozone trends in Japan and Mongolia were flat in the past decade. In the troposphere, the available measurements show an overall increasing tendency at different altitudes from a three-decade perspective and its trend in the past decade remains unclear due to data availability. The difference in tropospheric ozone level between East Asia and Southeast Asia is likely due to the high background ozone from stratospheric intrusion over Northeast Asia. In terms of ozone controls, our results suggest that anthropogenic emissions determine the occurrence of high ozone levels but the underappreciated strong ozone climate penalty, particularly over Southeast Asia, will make ozone controls harder under a warmer climate.

## 1. Introduction

Tropospheric ozone has been a long-lasting threat to public health, crop yield, and climate warming (Chang et al., 2017; DeLang et al., 2021; Lyu et al., 2023). Its importance in dampening carbon sink of forests by reducing productivity is also increasingly recognized in recent years (Cheesman et al., 2024). Tropospheric ozone is mainly produced from the photochemical reactions between nitrogen oxides ($NO_x$) and volatile organic compounds (VOCs) in the presence of sunlight, and stratosphere-troposphere exchange (STE) can also transport ozone into the troposphere (Neu et al., 2014) and even reach up to the surface under conducive weather conditions (Chen et al. 2024). In particular, high level of tropospheric ozone over East Asia and Southeast Asia is of great concern. The estimated cardiovascular premature mortality attributable to surface ozone is 277,800 (142,900-421,900) in 2019 over East Asia and Southeast Asia, accounting for ~50% of its global health burden (Sun et al., 2024). The current surface ozone exposure can reduce the annual crop yield in China, South Korea, and Japan, by ~60, 60, 20 million tonnes for wheat, rice, and maize, respectively (Feng et al., 2022). As such, it





is important to elucidate the spatiotemporal distributions of observed ozone from the surface to
troposphere over East Asia and Southeast Asia.
Surface ozone concentrations have been measured by the nation-level network for more than one
decade in many countries. In Japan, surface network since the 1970s revealed a gradual increase in
ozone (Nagashima et al. 2017; Kawano et al., 2022) until the past decade where Japanese sites
experienced an ozone decrease by -0.8±0.5 ppb yr$^{-1}$ (Wang et al., 2024). In South Korea, surface ozone
has been increasing in the past two decades, leading to the maximum daily 8 h average (MDA8) ozone
often exceeding 80 ppb in summer in the Seoul metropolitan area (Kim et al., 2023; Colombi et al.,
2023). In China, national surface network was established from 2013 and the widespread rising surface
ozone in the past decade positioned China to be one of countries with the highest ozone level
worldwide (Lu et al., 2020; Li et al., 2021; Wang et al., 2024). In contrast, Hong Kong, located in
China's southern coast, exhibited an overall increase in the surface ozone level by 0.35 ppb yr$^{-1}$ over
1994-2018, but the trend tended to level off in recent years (Wang et al., 2019).
In Southeast Asia, surface ozone levels are much smaller than those in East Asia due to the lower
anthropogenic emissions and frequent marine air inflow (Ahamad et al., 2020; Sukkhum et al., 2022;
Wang et al., 2022a). The previously published analyses on long-term ozone trends in Southeast Asia
are scarce, mainly focused on Malaysia and Thailand before 2016. In Malaysia, there was observed
ozone increase of 0.09-0.21 ppb yr$^{-1}$ over the Peninsular Malaysia during 1997-2016 but the Borneo
Malaysia recorded small or insignificant ozone trends (Ahamad et al., 2020; Wang et al., 2022a). In
Thailand, the observed surface ozone experienced significant increase by 0.7 to 1.2 ppb yr$^{-1}$ during dry
seasons over 2005-2016 (Wang et al., 2022a). In Indonesia, there was no significant ozone trend in
Bukit Koto Tabang (a suburban site) over 2005-2016 (Wang et al., 2022a). In Philippines, Salvador et
al. (2022) reported an increase of 0.41 ppb yr$^{-1}$ in surface ozone over 2014-2020 based on air quality
measurements in Butuan (an urban site), southern Philippines. Long-term ozone measurements in other
Southeast Asia countries were not well documented.
Tropospheric ozone profiles and columns over East Asia and Southeast Asia have been measured by
multiple platforms including ozonesonde, aircraft, and satellite. By using long-term ozonesonde
measurements, previous studies have extensively explored tropospheric ozone profiles in Beijing
(Zeng et al., 2023) and Hong Kong (Liao et al., 2020) of China, and in Pohang of South Korea (Bak
et al., 2022). However, these ozonesonde-based analyses mainly focused on the spatiotemporal
variability and source contributions of tropospheric ozone at the individual site. By using the IAGOS
(In-Service Aircraft for a Global Observing System) aircraft ozone observations, Gaudel et al. (2020)
show that tropospheric ozone level increases with latitude from Malaysia/Indonesia to Northeast





China/South Korea. More importantly, they reported a rapid tropospheric ozone increase in 1994–2016
over East Asia and Southeast Asia, consistent with satellite tropospheric ozone column trends
(Gopikrishnan and Kuttippurath, 2024), which has been further attributed to the rising anthropogenic
emissions both locally and remotely (Wang et al., 2022a; Wang et al., 2022b; Li et al., 2023).
Considering that East Asia and Southeast Asia has been identified as a global hot spot with the fastest
increase in observed tropospheric ozone after 1990s by the Intergovernmental Panel on Climate
Change (IPCC) Sixth Assessment Report (AR6), a comprehensive assessment on tropospheric ozone
over this region by using these available measurements is strongly needed.
Under the framework of the Tropospheric Ozone Assessment Report (TOAR, 2014-2019), the TOAR
documents comprehensively estimate the global ozone pollution and its historical trends. The first-
phase TOAR includes only limited ground observation data over East Asia and Southeast Asia
countries before 2014 (Chang et al., 2017). In the context of the TOAR Phase Two (TOAR II, 2020-
2024), the established East Asia Focus Working Group (EAWG) aims to advance ozone research over
East Asia and Southeast Asia, with a focus on observed ozone trends and their attributions. Please see
the accompanying paper for ozone trend attributions (Lu et al., 2024). Our effort is to include ozone
measurements (or post-calculated ozone metrics) from surface to tropopause collected from TOAR
database and individual institutions over East Asia and Southeast Asia.
This paper will present the most comprehensive view of ozone distributions and evolution over East
Asia and Southeast Asia across different spatiotemporal scales in the past two decades. The structure
of this paper is as follows: Section 2 introduces the multiple ozone measurements and calculation of
different ozone metrics; Section 3 describes the present-day surface ozone levels with different metrics
and long-term surface ozone trends in the past two decades; Section 4 describes the three-dimensional
present-day distribution and long-term trends in tropospheric ozone; Section 5 discusses the important
implications for future ozone pollution controls; Conclusions are given in Section 6.

**2. Data and methods**

**2.1 Surface ozone observations**

In this study, we used surface ozone measurements from national networks of China (2013-2021),
Japan (2005-2021), South Korea (2005-2021), Malaysia (2005-2021), and Thailand (2005-2021) that
were collected from the TOAR II database or provided by our EAWG members. In addition to the
national network records, individual ozone measurement in Ulaanbaatar of Mongolia, Phnom Penh of
Cambodia, and Bandung of Indonesia from the Acid Deposition Monitoring Network in East Asia



(EANET) was also included. To assess the long-term ozone trend in China before 2013, we also
collected 11 ozone measurements from previously-published literatures with updates from our EAWG
members. As shown in Table S1, it includes 1 global baseline station (Mt. Waliguan), 4 regional
background stations (Akedala, Longfengshan, Xianggelila, and Lin'an), and 1 rural station (Gucheng)
from Xu et al. (2020), 1 regional background station (Mt. Tai) from Sun et al. (2016), 1 regional
background station (Shangdianzi) from Ma et al. (2016), 1 urban station from Gu et al. (2020), and 1
urban station (Guangzhou) and 1 suburban station (Hong Kong) from Zhang et al. (2011).
To ensure data quality, the daily and monthly means were calculated using the hourly data when it has
over 75% valid data each day and month. To fully assess ozone distributions, we adopted the following
ozone metrics in this study: (1) Seasonal mean ozone. Seasonal MDA8 concentrations are calculated
for the four seasons (December-January-February, DJF; March-April-May, MAM; June-July-August,
JJA; September-October-November, SON), respectively. (2) Ozone exceedance. National ambient
ozone air quality standard varies greatly among countries in East Asia and Southeast Asia (Table S2).
The threshold for MDA8 ozone ranges from 60 μg m$^{-3}$ in Philippines to 160 μg m$^{-3}$ in China, and for
the maximum daily 1 h average (MDA1) ozone ranges from 120 μg m$^{-3}$ in Japan to 235 μg m$^{-3}$ in
Indonesia. Under standard conditions (1013 hPa, 273 K), 1 ppb = 2.14 μg m$^{-3}$. In this study, we adopted
the thresholds of 60 ppb and 47 ppb (WHO standard) for MDA8 ozone to determine the exceedance
days. (3) Peak season ozone. In 2021, the World Health Organization (WHO) newly introduced a
standard for the peak season (six-month mean) ozone limit of 60 μg m$^{-3}$ to save more people suffering
from its long-term exposure. We used this threshold to assessment the peak season ozone levels.

**2.2 Tropospheric ozone observations**

In this part, we suggest our results from the analysis of vertical ozone profile, mostly based on the
ozonesonde measurement and some aircraft measurement. There are a number of ozonesonde
measurement sites, but here, we only consider 10 sites (Table S3), which has 10 measurements per
year at minimum, and continues at least 5 years for enabling reliable characteristics. Data at 9 sites
were obtained from the World Ozone and Ultraviolet Radiation Data Centre (WOUDC), and data at
Beijing site was directly provided from Zhang et al. (2021).
We also used the altitudinal ozone measurements that have been collected from the In-service Aircraft
for a Global Observing System (IAGOS). While the IAGOS mission has been operational since 1990s
and still available, ozone data in East Asia are limited. Here we only utilized the IAGOS ozone data
from 1995 to 2014, the period having enough number of measurements. Location of all ozonesonde
sites and the IAGOS region are shown in Section 4.





**2.3 Ozone trend calculation**

In terms of ozone distributions, we present the present-day ozone maps averaged over 2017-2021. We required that there are at least three out of these five years of data available in the calculation. In terms of ozone trends: the time frame of 2013-2021 was adopted to represent the past decade trend; the time frame of 2005-2021 was adopted to represent the 21st Century trend and time series should begin at least in the range 2005-2010 and end in the range 2017-2021; the time frame of 1995-2021 was adopted to represent the late 20th century trend and time series should begin at least in the range 1995-1999 and end in the range 2017-2021.

Following TOAR II guideline, to determine the ozone trend, we first derived the monthly anomalies of ozone concentrations that are calculated as the difference between the individual monthly means and the monthly climatology. Then, a quantile regression method as recommended by TOAR II statistical guidance was employed to estimate the linear trend in surface ozone, and a 50th quantile regression slope was reported in consideration of the length of ozone records.

**3. Present-day distribution and long-term trends in surface ozone**

**3.1 Distribution of present-day surface ozone over 2017-2021**

**3.1.1 Seasonal mean MDA8 ozone**

Figure 1 shows the seasonal mean MDA8 ozone concentrations averaged over 2017-2021. In winter, seasonal mean ozone level is almost below 50 ppb and it is even decreased to 20-30 ppb in many Chinese cities. The high $NO_x$ emissions in urban environment make ozone strongly titrated and often drop below the Northern Hemisphere background ozone (Vingarzan, 2004). High ozone values of 55-60 ppb in Northern Thailand and 60-65 ppb in Bangkok (Thailand) are notable. In spring, seasonal mean ozone concentrations are doubled in North China (north of 30°N) and increased by 10-20 ppb from wintertime in South Korea and Japan. High ozone of over 60 ppb in Thailand still holds in spring and ozone concentration is enhanced by up to 20 ppb in Yunan province (China), reflecting a possible concentration from spring fire emissions over Southeast Asia (Xue et al., 2021). In summer, the highest ozone levels of over 75 ppb are found in the North China and western China exhibits ozone concentrations of 60-65 ppb. In Southern China, ozone level is decreased to 30~55 ppb because of the active summer monsoon rainfall (Zhou et al., 2022). The hot spot of summer ozone pollution is found in Seoul (South Korea) where seasonal mean ozone is also over 75 ppb, followed by 55~60 ppb in Tokyo (Japan), 40-50 ppb in Kuala Lumpur (Malaysia), 30-40 ppb in Bangkok (Thailand). In autumn, ozone concentrations are decreased strongly from their summer levels in the north of 30°N over East



Asia but are increased remarkably in the Pearl River Delta (PRD) region of China where its seasonal mean MDA8 ozone of up to 65 ppb is the highest level within the East Asia and Southeast Asia.

In addition to mean ozone level, Figure 2 shows the seasonal 95th percentile ozone concentrations averaged over 2017-2021. The ozone metric is almost the fifth highest value in each season, representing the high ozone values of great concern in air quality management. Although the seasonality of the 95th percentile ozone resembles the mean ozone evolution, the occurrence of the very high 95th percentile ozone values highlights the severity of ozone pollution over East Asia and Southeast Asia. In winter, high ozone of 85-95 ppb occurs over the Southern Thailand, and some cities in PRD region can suffer from ozone level over 75 ppb. In spring, in East Asia the 95th percentile ozone can reach over 95 ppb over Chinese major city clusters and Seoul, and in Southeast Asia ozone level of over 75 ppb occurs in many stations in Thailand and Peninsular Malaysia. In summer, high levels of the 95th percentile ozone appear exclusively over East Asia, with ozone concentrations of over 115 ppb in the North China Plain (NCP), over 105 ppb in the Yangtze River Delta (YRD), and over 95 ppb in PRD, Sichuan Basin, Seoul, and Busan. In addition, some cities (e.g., Tokyo, Osaka) in Japan also have ozone levels over 85 ppb. In autumn, the high ozone levels only concentrate on PRD and YRD regions, with the 95th percentile ozone over 115 ppb in PRD and over 95 ppb in YRD, respectively.

**3.1.2 Number of days of ozone exceedance**

Figure 3 shows that the national ozone air quality standard varies greatly in different countries over East Asia and Southeast Asia. For example, MDA8 and MDA1 ozone thresholds in China are 160 μg m$^{-3}$ and 200 μg m$^{-3}$, respectively, which lie at the high end of the adopted standards. A lower standard of MDA8 of 140 μg m$^{-3}$ in Thailand and of 120 μg m$^{-3}$ in Vietnam, South Korea, and Singapore are adopted, while Laos, Myanmar, and Philippine adopt a standard consistent with or lower than the WHO guidance. In terms of MDA1 standard, most of the countries adopt a threshold around 200 μg m$^{-3}$. As such, for the sake of health impact assessment, here we adopted the uniform threshold of 60 ppb and WHO guideline to estimate the annual ozone exceedance.

Figure 4 shows the annual number of days with MDA8 ozone concentration greater than 60 ppb (NDGT60) and with MDA8 ozone concentration greater than 47 ppb (NDGT47), respectively. In terms of NDGT60, most of the NCP cities in China have ozone exceedance over 125 days, followed by around 100 days in YRD, PRD, and Northwest China. In South Korea, most of the stations experience 60-100 days per year with daily MDA8 ozone over 60 ppb, while in Japan it is almost less than 45 days except for a few cities. In Southeast Asia, NDGT60 is almost less than 75 days, and particularly



Malaysia, Cambodia, and Indonesia have NDGT60 less than 15 days that is consistent with the very
low 95th percentile ozone (Figure 2). If the WHO standard is applied, most of the cities in eastern
China will have more than 150 days with MDA8 ozone exceedance, and this is also the case for western
China. This suggests the pressing challenge to mitigate ozone pollution due to the large-scale high
emissions in China. In South Korea, the NDGT47 is over 100 days for most of the stations, which is
consistent with the high background ozone issue as reported by Columbi et al. (2023). Ozone
exceedance over 100 days for NDGT47 can be also found in major cities in Japan, Thailand, and
Malaysia.

### 3.1.3 Peak season ozone levels

In this study, we also apply the new WHO standard for peak season ozone to assess risks of long-term
ozone exposure over East Asia and Southeast Asia. Figure 5 shows the estimated peak season ozone
concentrations averaged over 2017-2021 and its ratio relative to the WHO standard. In China, the NCP
region holds the highest peak season ozone of over 70 ppb that is about 2.5 times the WHO threshold,
followed by 65 ppb in YRD, 55 ppb in PRD, SCB, and some cities of Northwest China. More
importantly, the lowest peak season ozone in China is still higher than the WHO standard, suggesting
the difficulty in mitigation long-term ozone exposure over China. In South Korea, the peak season
ozone is well above 55 ppb and even higher than 60 ppb, again reflecting the important role of
background ozone in South Korea. In Japan, the peak season is mainly within the range from 40 to 55
ppb, amounting to 1.5-2 times the WHO standard. In Ulaanbaatar of Mongolia, the peak season ozone
is below 20 ppb. In Southeast Asia, Thailand has the highest peak season ozone of over 60 ppb around
Bangkok, and high values of 55-60 ppb are also found in the northern Thailand and southern coastal
Thailand. In Malaysia, the Peninsular Malaysia has peak season ozone of 30-50 ppb, higher than the
WHO standard. However, the Borneo Malaysia, Cambodia, and Indonesia record peak season ozone
lower than the WHO standard. Overall, the estimated peak season ozone level shows that 98% stations
in East Asia and Southeast Asia are above the WHO standard, and suggests the urgent need to reduce
long-term ozone exposure risks.

### 3.2 Surface ozone trends in the past two decades

### 3.2.1 2005-2021 ozone trends

Figure 6 shows the observed ozone trends in different seasons over the period of 2005-2021. Due to
the availability of long-term surface measurements, we only present ozone trends over South Korea,
Japan, Thailand, and Malaysia. In South Korea, increasing ozone trends with high certainty are notable
across different seasons ranging from 0.48 ppb yr$^{-1}$ in winter to 0.96 ppb yr$^{-1}$ in summer. In Japan,



observed ozone shows a decreasing tendency from 2005 to 2021 in summer but an extensive ozone
increase by 0.28 ppb yr⁻¹ in wintertime. In Thailand, there is an overall increasing trend in surface
ozone but with spatial heterogeneity over 2005-2021. Specifically, significant ozone increase mainly
occurs over northern Thailand and southern coastal Thailand, while ozone increase around Bangkok
is much smaller or insignificant. In Malaysia, there is a wintertime ozone increase by 0.2 ppb yr⁻¹
particularly in three sites in Peninsular Malaysia and in five sites in Borneo Malaysia, while in other
seasons the observed ozone trends over 2005-2021 are small and statistically insignificant. The
estimated increasing tendency in surface ozone since 2005 is in agreement with Kim et al (2023) for
2001-2021 ozone increase in South Korea and with Wang et al. (2022) for 2005-2016 ozone increase
in Southeast Asia.
Due to the lack of national network measurement before 2013 in China, we also complied 11 individual
ozone measurements (8 background/rural sites and 3 urban sites) that are available from around 2005
(see Data and methods). Figure 7 and Table S1 show the estimated seasonal ozone trends in these 11
stations by using the metrics of MDA8 ozone and 24-hour mean ozone. The Mt. Waliguan, a global
baseline station of the World Meteorological Organization /Global Atmosphere Watch (Xu et al., 2020),
shows statistically significant ozone increase by 0.56 ppb yr⁻¹ in spring. However, at the multiple
regional background stations located in western boundary of China (Xianggelila, Akedala) and eastern
boundary of China (Lin'an, Longfengshan), there is no such a consistent ozone increase but with large
variability across different seasons, suggesting the important role of regional emission change and
climate variability (Zhang et al. 2023, Ye et al., 2024). In the NCP, one of the regions with the highest
present-day ozone level, the observed ozone after 2005 at the regional background sites (Shangdianzi,
Mt. Tai) and rural site (Gucheng) experienced a consistently increasing trend in spring and summer
seasons. In Shangdianzi, the MDA8 ozone trend over 2005-2019 is 0.85 ppb yr⁻¹ (p<0.1) in spring and
0.73 ppb yr⁻¹ (p=0.12) in summer, respectively. The similar seasonal trends are also shown in Gucheng
(a rural site close to Shangdianzi) and Mt. Tai (located in the center of NCP). It is noted that summer
ozone trends in Mt. Tai over 2005-2019 also have strong intraseasonal variability, with much faster
ozone increase in July and August (Sun et al., 2016). In addition to the background/rural sites, urban
sites in YRD (Xujiahui) and PRD (Guangzhou, Hong Kong) record the urban ozone increase after
2005 that has been attributed to anthropogenic emissions and circulation patterns in previous studies
(Wang et al., 2019; Gu et al., 2020; Cao et al., 2024).

**3.2.2 2013-2021 ozone trends**

Figure 8 shows the observed ozone trends in different seasons over the period of 2013-2021. Here we
include ozone trends over China, Mongolia, Japan, South Korea, Malaysia, and Thailand. In China,



there is a widespread ozone increase throughout the year, with mean ozone increase of 1.0-1.2 ppb yr⁻
[1] in different seasons, which is only half of the ozone increase over 2013-2019 in China (Lu et al.,
2020; Li et al., 2020). Spatially, ozone increase mainly occurs in the northern China and western China.
Seasonally, there is fast ozone increase in winter over the NCP region, suggesting the urgency of
wintertime ozone regulation (Li et al., 2021). In South Korea, the 2005-2021 ozone rise is strongly
mitigated over 2013-2021 when summer ozone trend is only 0.45 ppb yr⁻¹. In Mongolia, there is a
notable spring ozone increase but with low certainty. In Southeast Asia, however, the observed ozone
in Malaysia and Thailand shows a decreasing tendency in most of the sites, which is contrary to the
over ozone increase from 2005 to 2021. Overall, except for the rapid ozone increase over China in the
past decade, there is a leveling off or decrease in surface ozone trend over other countries in the
meantime.
To further examine the long-term ozone variability, we also show the time series of observed national
MDA8 ozone concentrations during warm seasons from 2005 to 2021 in Figure 9. In South Korea,
there is a flat trend in ozone over 2017-2021 after a sustained ozone increase since 2015, and there is
no clear trend in warm-season ozone in Japan due to the limited data availability. In Southeast Asia,
after 2013, surface ozone in Malaysia starts to decline and ozone trend in Thailand levels off. This is
also demonstrated in the warm-season ozone trend in Figure S1. In addition, we also find the large
interannual variability in observed ozone concentration that deserves further investigation. For
example, in 2017, there is strong surface ozone enhancement relative to 2016 in China, Japan, and
South Korea, while surface ozone is consistently decreased in Mongolia, Thailand, and Malaysia.
Previous studies have linked the changes in large-scale circulations to this extensive ozone anomalies
(e.g., Yin et al., 2010; Jiang et al., 2021).

**4. Present-day distribution and long-term trends in tropospheric ozone**

**4.1 Three-dimensional distribution of present-day tropospheric ozone**

First, we compared climatological mean vertical ozone profile (from surface to 10 km altitude) using
the ozonesonde data (Figure 10). Beijing site in China shows the highest, but Sepang Jaya site in
Malaysia shows the lowest ozone mixing ratio through the troposphere. In general, ozone mixing ratio
in East Asia (Beijing in China, Pohang in Korea, and Tsukuba in Japan) is higher than that in Southeast
Asia (Sapang Jaya in Malaysia and Watukosek in Indonesia). This pattern is well found when we
compared average ozone mixing ratio at 1, 3, 5, and 7 km altitude (Figure 11). While some sites show





the higher ozone mixing ratio in the boundary layer (e.g., Watukosek), but generally free tropospheric
(above 1-2 km height) ozone mixing ratio is higher. Especially, Beijing, Pohang, Sapporo, and Tsukuba
sites show large enhancement of ozone above 8 km altitude (Figure 11a), implying that the
stratospheric ozone is used to be strongly intruded into the troposphere. Actually ozone mixing ratio
values in these 4 sites are highest at 3, 5, and 7 km altitudes, indicating the effect of stratospheric ozone
to the enhancement of tropospheric ozone. These 4 sites are located over the Korean peninsula (Figure
10) where sudden increase of ozone usually occurs below the tropopause (Park et al., 2012).
Seasonal pattern of vertical ozone profile was continually investigated (Figure 12). Tropospheric ozone
values at Beijing, Pohang, Sapporo, and Tsukuba site where strong stratospheric ozone intrusion occurs,
are generally high in spring (MAM) and summer (JJA). This pattern can be explained by the frequent
intrusion of stratospheric ozone in spring (Park et al., 2012), and strong photochemical ozone
production that is typical characteristic in summer. In several sites (e.g., Beijing and Tsukuba),
photochemical ozone production in summer makes the boundary layer ozone much higher than free-
tropospheric ozone. Stratospheric ozone intrusion in these 4 sites is also strong in winter, but does not
result in high tropospheric ozone due to weak photochemistry in winter. Ozone values at Kagoshima
(Japan), Naha (Japan), King's park (Hongkong), and Hanoi (Vietnam) that are located below 30 °N,
however, are lowest in the lower troposphere. Considering that these sites are easily affected by the
inflow of maritime air mass under the trade-wind influence, this low summertime ozone can be
explained by the transport of humid and ozone-poor air mass from the ocean due to the monsoon
system (Zhao and Wang, 2018; Jiang et al., 2021). Sites in equatorial region (i.e., Sepang Jaya and
Watukosek) do not have large seasonal variation of tropospheric ozone.
We repeated same analysis using the IAGOS data (Figure 13). IAGOS ozone profiles over Northeast
Asia also reveal the highest tropospheric ozone in summer (June), and lowest in winter (December).
We can also see large enhancement of summertime ozone in the boundary layer associated with strong
photochemistry, and highest ozone in winter (DJF) and spring (MAM) above 8 km altitude, implying
the intrusion of stratospheric ozone. Monthly variation of ozone at multiple heights (Figure 13b)
illustrates a sharp drop of ozone from June to July, depicting the wash-out effect due to the rainy season
called Jangma (Korea) or Maiyu (China). Overall, ozone profile pattern in Northeast Asia from the
long-term aircraft monitoring is similar to findings based on ozonesonde measurements. Among them,

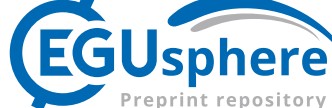

we would highlight that the site showing high tropospheric ozone (e.g., Beijing in China, Pohang in
Korea, Sapporo in Japan), which are located in Northeast Asia and latitude is higher than 35 °N (Table
S3), relate to the strong intrusion of stratospheric ozone. Considering recent studies addressing that
background ozone in Northeast Asia is unexpectedly high (Lee and Park, 2022; Columbi et al., 2023),
we need to put more weight on the study about the contribution of stratospheric air masses to the
Northeast Asian background ozone. Also, some previous studies reported cases of the tropospheric
ozone enhancement in Southern China affected by the influence of typhoon (Zhan and Xie, 2022; Li,
F. et al., 2023), which are typically explained based on the stratospheric ozone intrusion driven by the
deep convection (Chen et al., 2022). While those reported cases look significant, however, our results
in sites typically affected by typhoon (e.g., Naha, King's park) reveal that it may not contribute to
significant increase of summertime mean tropospheric ozone. We also added analyzed results using
the IAGOS measurements in Southeast Asia, but the measurements were performed in some limited
periods. There is no available data after 2012, and the number of data is enough to analyze only for the
year 1995, 1996, 1997, 1999, and 2005. Thus, we did not deeply interpret IAGOS results in Southeast
Asia, but simply reported themselves.
**4.2. Altitudinal long-term trends of tropospheric ozone**
In addition to the spatial distribution of tropospheric ozone, we investigate the long-term trend of ozone
mixing ratio in a vertical scale using the ozonesonde measurements. We confirmed the time-series
analysis at each altitude (Figure S2) and performed the Mann-Kendall test. Finally, we estimated long-
term ozone trend in the troposphere (from surface to 10 km altitude) per 100 m interval vertically with
the information of statistical significance. These results are shown in Figure 14.
At first, we can see increasing trend of tropospheric ozone in some East Asian sites that we are treating.
Increasing trend of ozone mixing ratio about 1-2% per year is found at Sapporo, Naha, and Hanoi
consistently through whole troposphere (Figure 14a, 14e, and 14g). Tsukuba and Pohang sites have
similar pattern but smaller trend (~0.5-1 % per year). Ozone in King's park, Sepang Jaya, and
Watukosek are only increasing in the boundary layer (below ~2-3 km), but almost no significant long-
term trend in the free troposphere. Kagoshima and Beijing sites are totally opposite; There are
decreasing trends through whole troposphere. In brief, we can classify 3 types of long-term trends of
tropospheric ozone in East Asia: (1) Increase through whole troposphere, (2) Increase only in the





boundary layer and no clear trend in the free troposphere, and (3) Decrease through whole troposphere.
We also examined trends using the seasonal mean ozone mixing ratio: MAM in Figure S3, JJA in
Figure S4, SON in Figure S5, and DJF in Figure S6. Overall, we can split two different patterns such
as seasonally consistent and inconsistent trends. Tropospheric ozone at Sapporo, Tsukuba, and Naha
has been consistently increasing trends in all seasons. In contrast, Tropospheric ozone at Beijing
reveals consistent decreasing trend, only with some exception. Some exceptions are increasing trends
near the surface in DJF and MAM. While these are not statistically significant, it seems required to put
our eyes here more because near-surface ozone increase in high polluted area directly connects to the
human health and crop damage. We can state that tropospheric ozone trend at King's park (increasing),
Hanoi (increasing), and Kagoshima (decreasing) is rather consistent in all seasons, but the extent of
trend varies largely according to the season. Trends at Pohang, Sepang Jaya, Watukosek are seasonally
different. Ozone trends at Pohang are clearly positive in JJA and SON but almost none or even partly
negative in upper heights in DJF and MAM Trends at Sepang Jaya are only positive in DJF, but
generally none or negative in other seasons. Ozone at Watukosek shows the distinguished increasing
only in MAM. These features imply that a certain season has matchless trend value and it can lead
whole trend pattern in that site.
We finally estimated the long-term trend of tropospheric ozone in East Asia using the IAGOS aircraft
measurements (Figure S7). Data is only available from 1995 to 2014, therefore recent decade situation
(e.g., the outbreak of Coronavirus disease 2019) is not included here. In spite of this limitation,
generally we can see the increasing trend of tropospheric ozone in East Asia, consistent with previous
reports (Wang et al., 2019; Lee et al., 2021; Li, S. et al., 2023). Seasonally, however, trends are rather
different; There are clear increasing trends during JJA and SON, but almost no trend with partial
decreasing trend in the upper troposphere during DJF and MAM. Partial decreasing trends in DJF and
MAM look similar to a recent report addressing that stratospheric ozone transport to the troposphere
in has been weakened (Chen et al., 2024), but overall, tropospheric ozone in East Asia reveals large
increasing trends in warm season (JJA and SON), and it seems to lead to an overall ozone increase in
East Asia.



**5. Implications for ozone control**

Our research reveals significant spatial and seasonal ozone variations over East Asia and Southeast Asia. Spatially, ozone levels are closely associated with anthropogenic emissions (e.g., $NO_x$ emissions), with high ozone concentrations aligning well with $NO_x$ emission patterns observed through ground-based and satellite measurements. Figure 15 shows the bottom-up $NO_x$ emissions and the satellite-derived $NO_2$ columns over East Asia and Southeast Asia. Seasonally, ozone variations are primarily influenced by meteorological conditions and biomass burning emissions in Southeast Asia. For example, ozone peaks usually occur in northern China during summer, in the Pearl River Delta during autumn, and in Southeast Asia during spring.

Relative to East Asia, although the health risks in Southeast Asia are relatively low under short-term ozone exposure indicators (e.g., 95th percentile ozone concentration), the WHO newly introduced peak season ozone concentration standard indicates that both East Asia and Southeast Asia are faced with a widespread risk of long-term ozone exposure, with the vast majority of the region exceeding WHO standards. In addition to health impacts, the pervasive ozone pollution in East Asia and Southeast Asia is also threatening global food security by its accounting for over 60% of global rice yield (Feng et al. 2022; Yuan et al., 2022). For example, the year-around mean MDA8 ozone over 40 ppb over Southeast Asia suggests the high ozone exposure over a threshold of 40 ppb (AOT40) that is commonly used to investigate ozone effects on vegetation yield (Feng et al. 2022).

In addition to the well-known fast-changing anthropogenic emissions over East Asia (Zheng et al., 2018) and Southeast Asia (Wang et al., 2022), our study shows that there is a very strong ozone climate penalty over East Asia and Southeast Asia. Figure 16 shows the observed 50th percentile regression slope between MDA8 ozone and temperature in different seasons averaged over 2017-2021. In East Asia, the locations of high ozone-temperature slope of 3-5 ppb °C$^{-1}$ in different seasons are consistent with the observed high level of surface ozone. The highest slope of over 5-8 ppb °C$^{-1}$ is found over the PRD and Sichuan Basin in summer. In Southeast Asia, however, we find a widespread high ozone-temperature slope. In Thailand, the ozone-temperature slope of over 3 ppb °C$^{-1}$ can be found throughout the year expect for summer. In Malaysia, a strong slope of 4-8 ppb °C$^{-1}$ persists all the year around that is consistent with a ten-year analysis in Kuala Lumpur by Ashfold et al. (2024). More importantly, the observed 95th percentile regression shows a notably increased ozone-temperature slope over Southeast Asia (Figure S8), suggesting a stronger ozone climate penalty under extreme conditions. In contrast, the IPCC AR6 only identified East Asia and India as the hotspot of ozone climate penalty (Zanis et al., 2022). Our observed-based results highlight the strongly underestimated ozone climate penalty over Southeast Asia.



The long-term trend of surface ozone indicates that, based on the available data, high-emission regions
in South Korea, Southeast Asia, and China have generally experienced an increase in ozone levels
since 2005. However, since 2013, the increase in ozone levels in China has significantly accelerated,
while the ozone trends in Thailand and Malaysia in Southeast Asia show no significant changes.
Therefore, it is still urgent to attribute the varying ozone trends in East Asia and Southeast Asia across
different seasons over the past decade.
In the troposphere, the available ozonesonde and IAGOS measurements not only demonstrate the high
background ozone in warm seasons over Northeast Asia, but also show an overall increasing tendency
in the past three decades. While the increase in tropospheric ozone can be largely attributed to the
increased anthropogenic emissions as demonstrated in our companion paper (Lu et al., 2024), the
origin of high seasonal background ozone in Northeast Asia remains unclear. Recent studies provide
some observational and modeling evidence of stratospheric intrusion (Chen et al., 2024; Columbi et
al., 2023) to explain this high background ozone, but a quantitative assessment is urgently needed. In
particular, the recent ASIA-AQ campaign (https://espo.nasa.gov/asia-aq) flying across Asia counties
would be important to understand the high tropospheric ozone issue over East Asia and Southeast Asia.

## 6. Conclusions


Under the framework of the TOAR II (2020-2024) that aims to estimate global and regional
tropospheric ozone pollution and its historical trend, in this study we present the most comprehensive
view of ozone distributions and evolution over East Asia and Southeast Asia across different
spatiotemporal scales in the past two decades. This is done by taking advantage of the available surface
ozone measurement in the past two decades (2005-2021) over eight countries, and ten ozonesonde and
in-service aircraft measurements within this region. The key conclusions are as follows:
Firstly, there are significant spatial and seasonal ozone variations at the present-day level. In summer,
seasonal mean MDA8 ozone averaged over 2017-2021 varies from 30 ppb in Southeast Asia to over
75 ppb in summer in North China and Seoul. Southeast Asia in winter and spring has high mean ozone
of 60 ppb in Thailand. The seasonality of the 95th percentile ozone resembles the mean ozone evolution,
but the widespread occurrence of the very high 95th percentile ozone of over 85 ppb highlights the
severity of ozone pollution. If the WHO standard is applied for short-term exposure, a large fraction
the sites will have more than 100 days with MDA8 ozone exceedance. In terms of long-term exposure,
the WHO newly-introduced peak season ozone standard indicates that both East Asia and Southeast

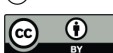



Asia are faced with a widespread risk of long-term ozone exposure.
Secondly, the surface ozone increase in the past two decades is widespread. In particular, South Korea
has a national ozone increase with high certainty across different seasons. In Thailand, there is an
overall increasing trend in surface ozone but with spatial heterogeneity over 2005-2021. In China, the
complied 11 individual measurements show an overall ozone increase in high-emission regions and at
a global baseline station. However, the observed national surface ozone increase in South Korea and
Southeast Asia from 2005 is leveling off or even decreased in the past decade (2013-2021), while
ozone increase in 2000s over China has amplified after 2013. Surface ozone trends in Japan and
Mongolia are generally flat in the past decade.
Thirdly, in the troposphere, the high ozone levels in spring and summer at Beijing, Pohang, Sapporo,
and Tsukuba site are driven by strong photochemical ozone production and stratospheric ozone
intrusion, supported by both the ozonesonde and IAGOS measurements. The difference in tropospheric
ozone level between East Asia and Southeast Asia is likely due to the high background ozone from
stratospheric intrusion over Northeast Asia. In terms of ozone trends, from a three-decade perspective,
the available ozonesonde and aircraft measurements show an overall increasing tendency at different
altitudes but feature with strong site-by-site differences. Due to measurement availability, ozone trend
in the past decade is still unquantified.
Fourthly, the significant spatial ozone variations over East Asia and Southeast Asia are closely
associated with anthropogenic emissions, supported by ground-based and satellite measurements. Our
study also shows that there is a very high ozone climate penalty over East Asia and Southeast Asia,
and the widespread high ozone-temperature slope of 3-8 ppb $°C^{-1}$ persists all the year around in
Southeast Asia. More importantly, the observed 95th percentile regression shows a notably increased
ozone-temperature slope over Southeast Asia, suggesting a critical issue in future ozone controls.








**Data availability**. All data used in this study, including the observations, meteorological reanalysis and emission data will be archived in a freely-accessed data portal upon the publication of the manuscript.

**Supplement.** The supplement related to this article is available online.

**Author contributions.** K.L. and J.H.K. led and organized the project, working as the co-leads of the East Asia Working Group of Tropospheric Ozone Assessment Report Phase II (TOAR II) with X.L. and T.N. S.J.F., J.Q.Z., X.P.L., L.K.X., J.M.X., Y.X.G., Z.Q.M., M.T.L., T.A., J.G., M.H.L., J.B., J.K., J.H.K., X.L., and T.N. assisted in preparation of observational data. K.L., R.T., W.H.Q., and J.H.K. conducted the analysis and prepared the figures. T.L., Y.F.W., D.Y.T.Z., M.L.T., W.Q.Z. contributed to preparing the figures. K.L. and J.H.K. wrote the draft with inputs from H.L. All authors contributed to improving the manuscript.

**Competing interests.** Some authors are members of the editorial board of Atmospheric Chemistry and Physics.

**Acknowledgements**. We greatly thank the China's Ministry of Ecology and Environment, Korean Ministry of Environment, National Institute for Environmental Studies, Malaysia Department of Environment, Thailand Pollution Control Department, the Acid Deposition Monitoring Network in East Asia (EANET), World Ozone and Ultraviolet Radiation Data Centre (WOUDC), and In-service Aircraft for a Global Observing System (IAGOS) for running the ozone observation networks. We also thank the previous and current TOAR Steering Committee members (Owen Cooper, Lin Zhang, and Keding Lu) for effortless support of guiding the East Asia Working Group of Tropospheric Ozone Assessment Report Phase II (TOAR II).

**Financial support**. This research was supported by the National Natural Science Foundation of China (grants 42293323, 42205114, and 42293321), the Natural Science Foundation of Jiangsu Province (BK20240035). This work was also supported by the National Research Foundation of Korea (NRF) grant funded by the Korea government (MSIT) (RS-2023-00219830).



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





**Figures**

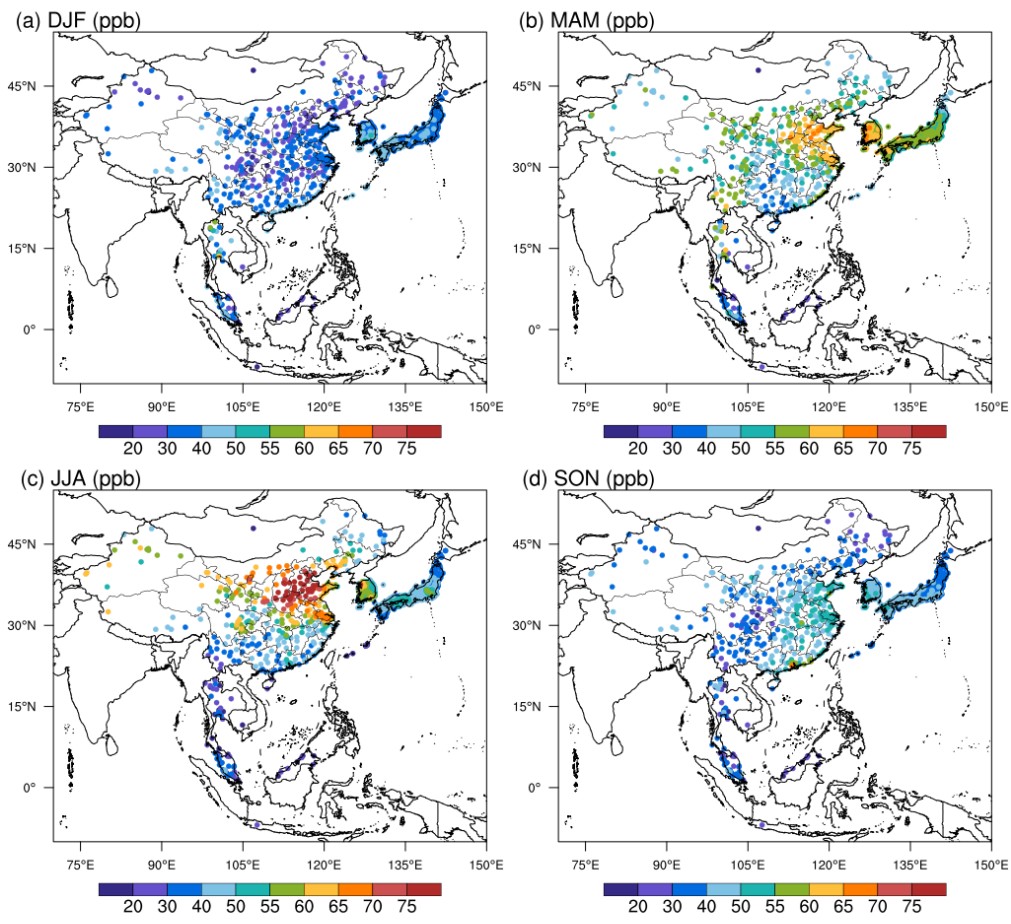


**Figure 1.** The observed seasonal mean MDA8 ozone (ppb) in (a) DJF, (b) MAM, (c) JJA, and (d) SON
averaged during 2017-2021 over East Asia and Southeast Asia. There are eight countries with surface
ozone measurements, including Cambodia (1 site), China (360 sites), Indonesia (1 site), Japan (1187
sites), Malaysia (66 sites), Mongolia (1 site), South Korea (473 sites), and Thailand (25 sites).



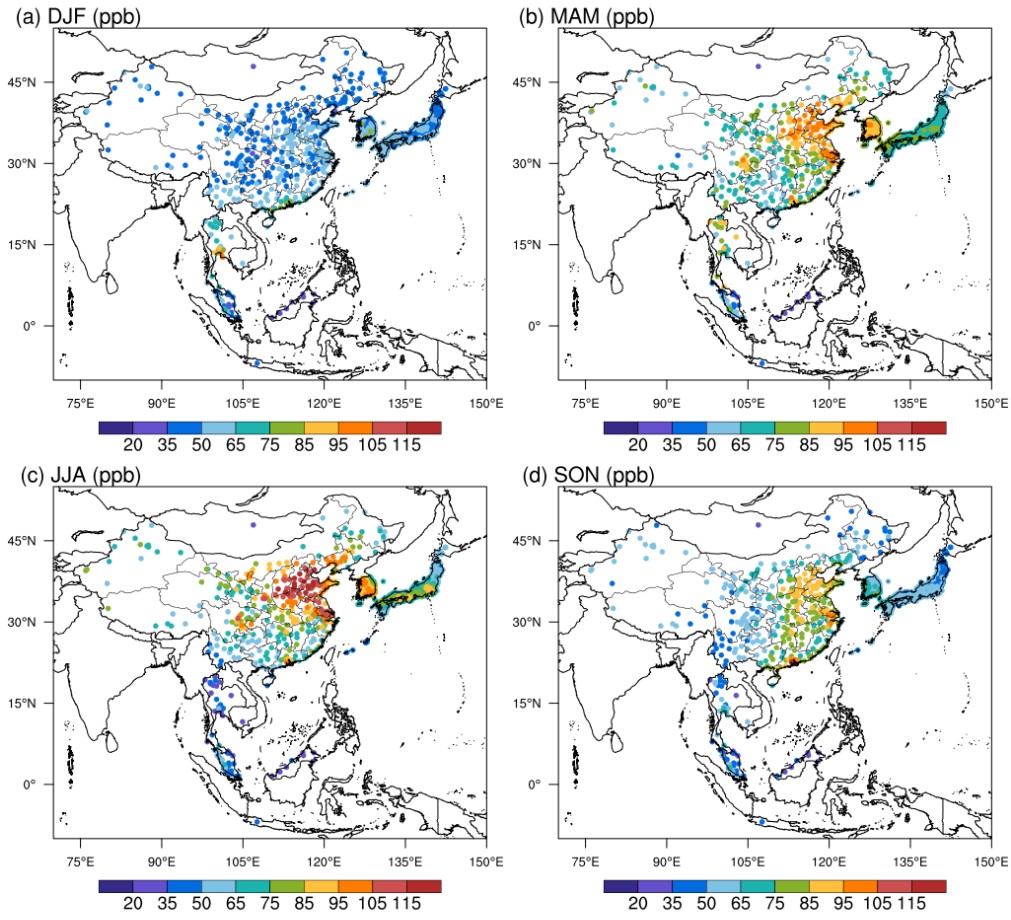

**Figure 2.** Same as Figure 1 but for the seasonal 95th percentile MDA8 ozone (ppb) averaged over 2017-2021. This metric represents the extreme high ozone values that are related to short-term ozone exposure.



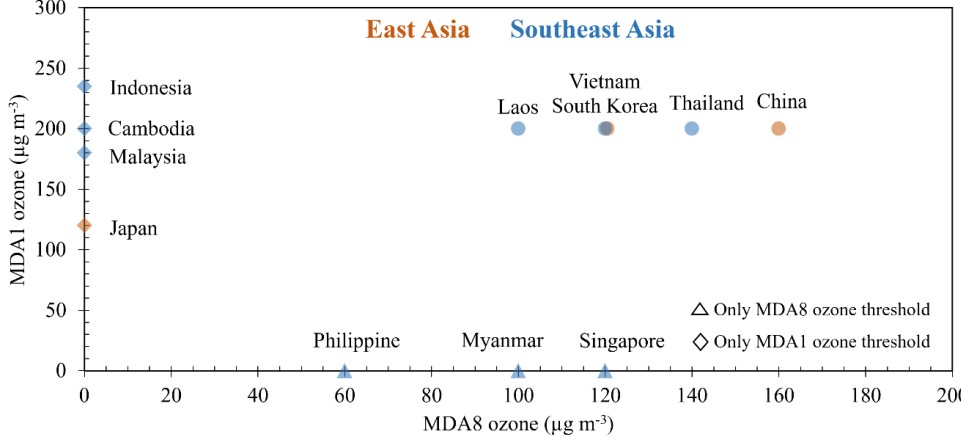

**Figure 3.** The national ambient ozone air quality standard in East Asia and Southeast Asia. The maximum daily 8 h average (MDA8) and/or maximum daily 1 h average (MDA1) ozone thresholds are routinely adopted but they vary greatly in different countries. The sources for these thresholds are given in Table S1. Under standard conditions (1013 hPa, 273 K), 1 ppb = 2.14 µg m$^{-3}$.



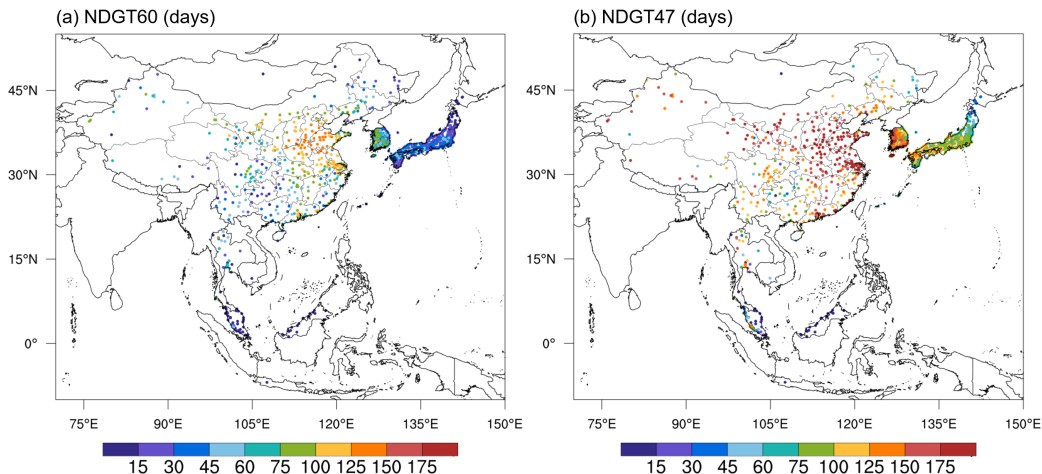

**Figure 4.** Annual number of days with daily MDA8 ozone greater than 60 ppb (NDGT60) and greater than the WHO standard of 100 µg m$^{-3}$ (NDGT47) averaged over 2017-2021. Under standard conditions (1013 hPa, 273 K), 1 ppb = 2.14 µg m$^{-3}$.




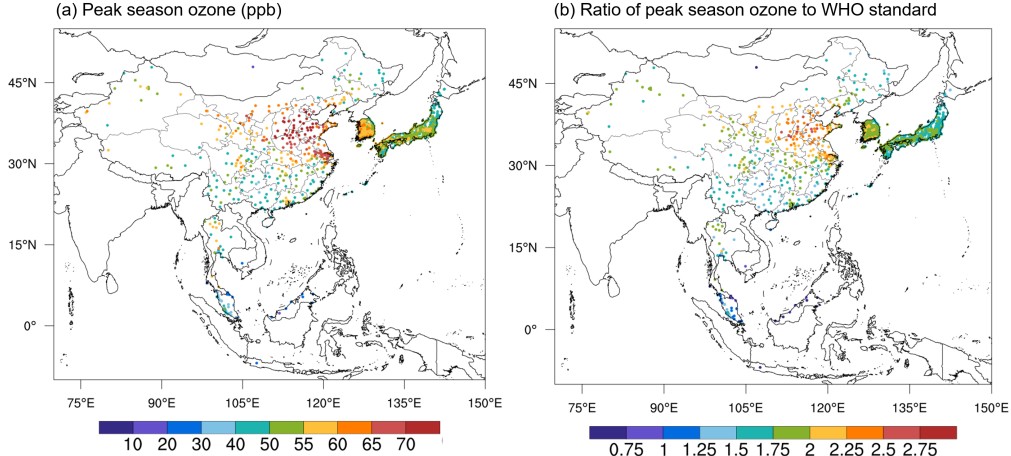

**Figure 5.** Annual mean peak season ozone (ppb) averaged over 2017-2021 (a) and the ratio of the observed peak season ozone to the WHO standard of 60 μg m$^{-3}$ (b). As introduced by the WHO, the concentration of peak season ozone is calculated by using the average monthly MDA8 ozone concentration in the six consecutive months with the highest six-month running-average ozone concentration. This new metric represents the long-term ozone exposure.

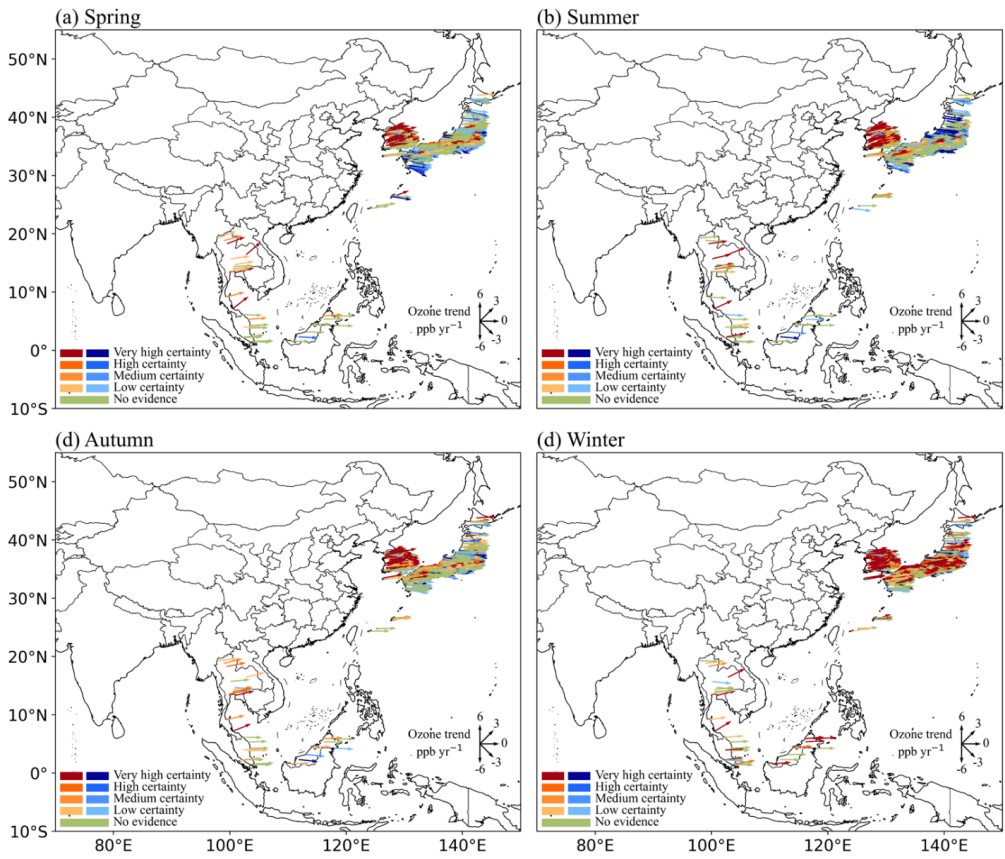

810

**Figure 6.** The observed 2005-2021 ozone trends (ppb yr$^{-1}$) during (a) spring, (b) summer, (c) autumn, and (d) winter over East Asia and Southeast Asia. Here it only includes ozone measurements from Malaysia (19 sites), Japan (946 sites), South Korea (226 sites), and Thailand (13 sites). National surface ozone data in China is not available before 2013, therefore not shown in this figure. To follow the trend reliability scale recommended by the TOAR II, here we use "very high certainty" to denote $p \leq 0.01$, "high certainty" to denote $0.05 \geq p > 0.01$, and "medium certainty" to denote $0.10 \geq p > 0.05$; positive trends are in red and negative trends are in blue.



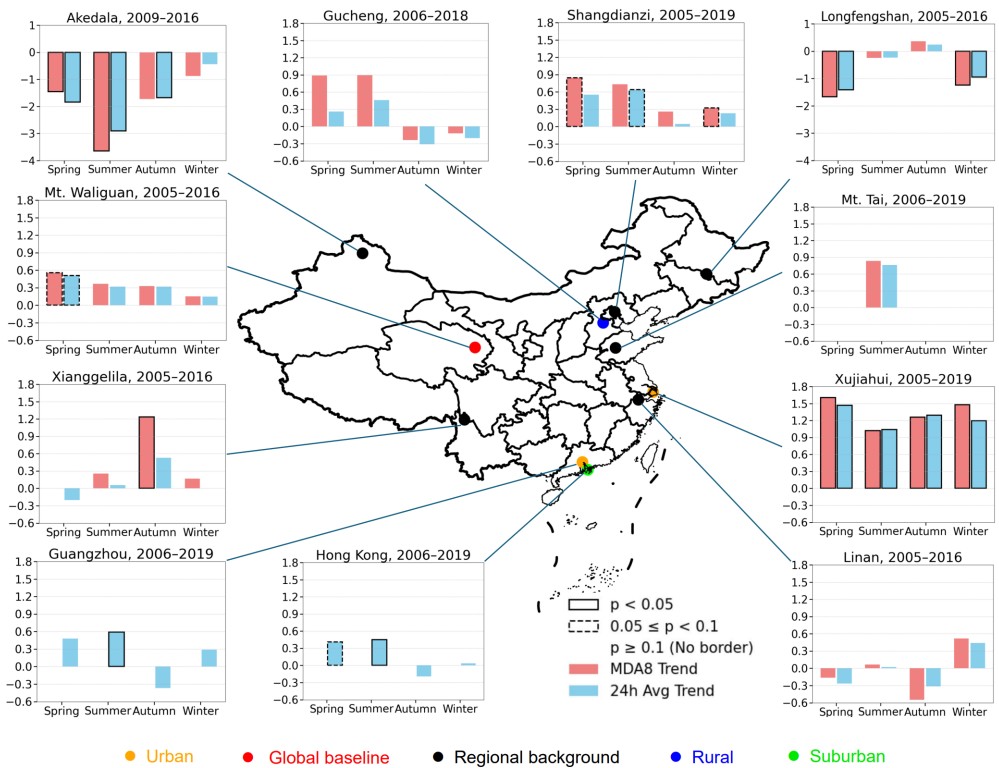

**Figure 7.** The observed long-term ozone trends after 2005 in 11 measurement sites over China. There are 1 global baseline station, 5 regional background stations, 1 rural station, 1 suburban station, and 2 urban stations. Due to data availability, we use the MDA8 ozone and/or 24-hour mean ozone in the calculation of ozone trends. The *p*-value for estimated ozone trends is also highlighted by rectangles.

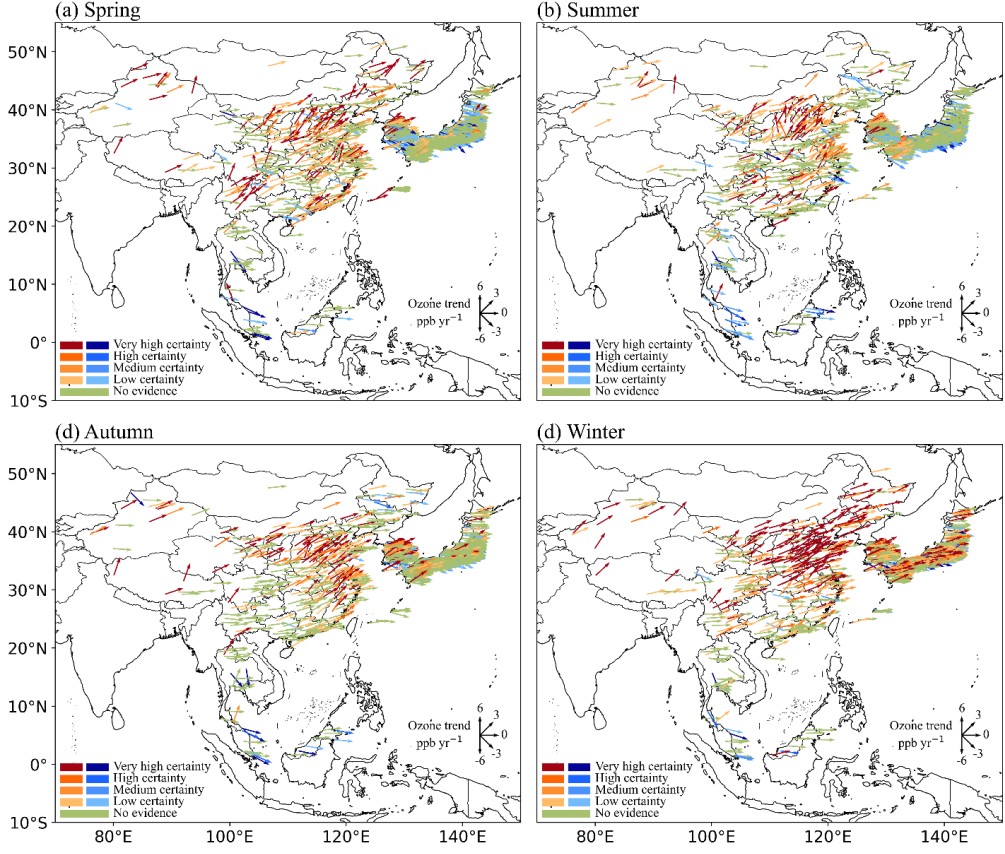

**Figure 8.** Same with Figure 6 but for the observed 2013-2021 ozone trends (ppb yr⁻¹) over East Asia and Southeast Asia. Here it includes ozone measurements from China (335 sites), Malaysia (19 sites), Mongolia (1 site), Japan (1130 sites), South Korea (270 sites), and Thailand (22 sites). To follow the trend reliability scale recommended by the TOAR II, here we use "very high certainty" to denote $p \leq 0.01$, "high certainty" to denote $0.05 \geq p > 0.01$, and "medium certainty" to denote $0.10 \geq p > 0.05$; positive trends are in red and negative trends are in blue.




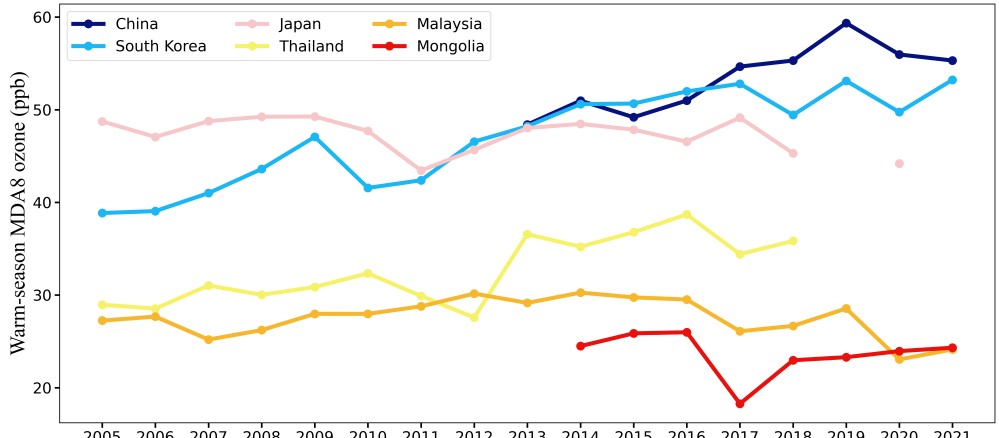


**Figure 9.** The observed national mean MDA8 ozone (ppb) during warm seasons (April to September)
from 2005 to 2021 in East Asia and Southeast Asia.




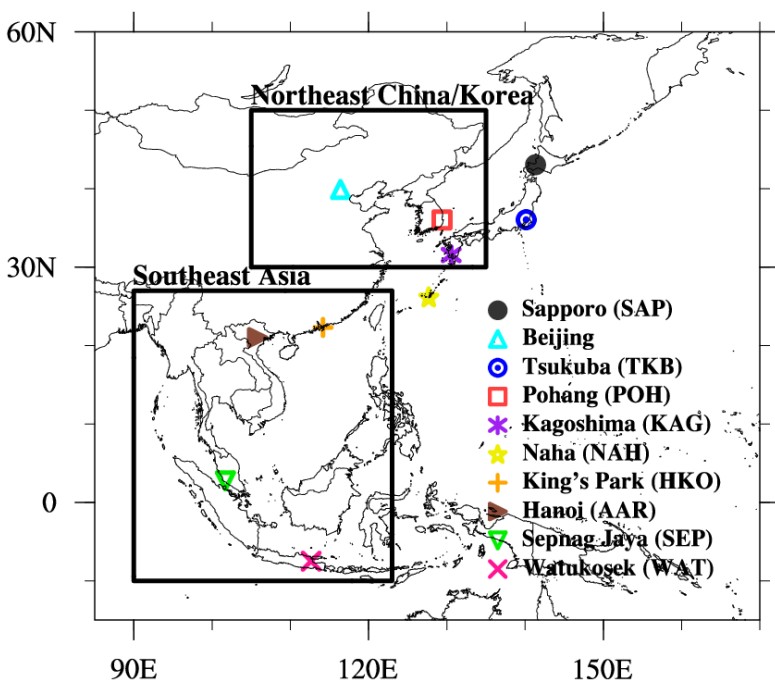

**Figure 10**. Map showing the location of ozonesonde sites and the coverage of the IAGOS
measurements considered in this study.



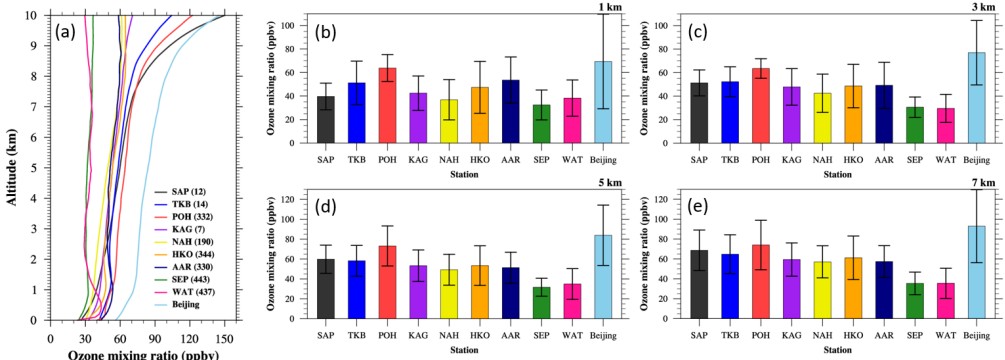

**Figure 11**. (a) Climatological mean vertical ozone profiles of 10 ozonesonde sites in the troposphere (from 0 to 10 km altitude) are compared. Also, mean ozone mixing ratio values of 10 ozonesonde sites at (b) 1 km, (c) 3 km, (d) 5 km, and (e) 7 km altitude are compared. Error-bar shows the 1-sigma standard deviation range.





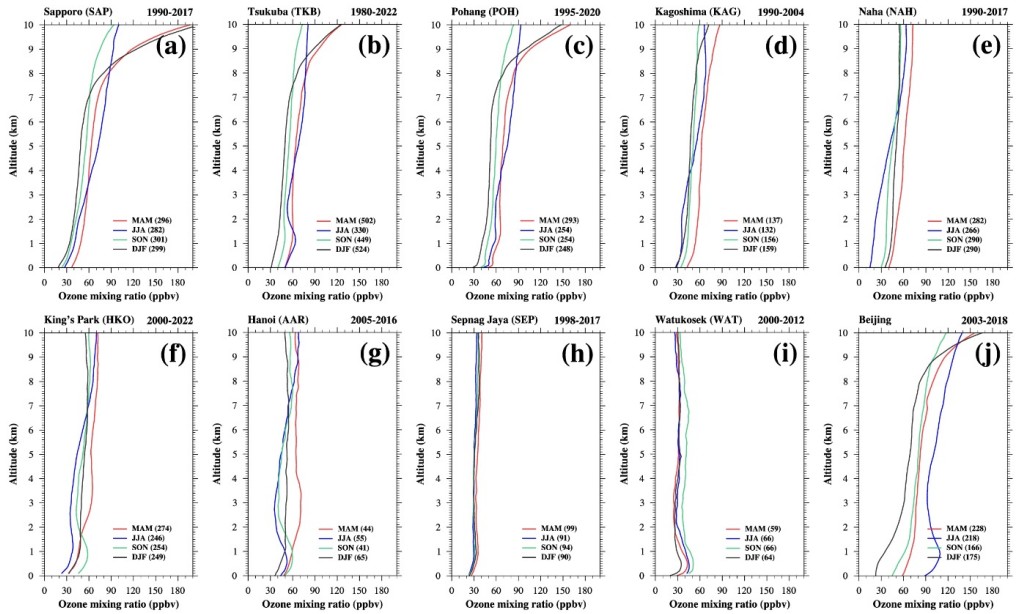

**Figure 12**. Seasonal mean vertical ozone profiles at (a) Sapporo (SAP), (b) Tsukuba (TKB), (c) Pohang
(POH), (d) Kagoshima (KAG), (e) Naha (NAH), (f) King's park (HKO), (g) Hanoi (AAR), (h) Sepang
Jaya (SEP), (i) Watukosek, and (j) Beijing site: March-April-May (MAM, red), June-July-August (JJA,
blue), September-October-November (SON, green), and December-January-February (DJF).



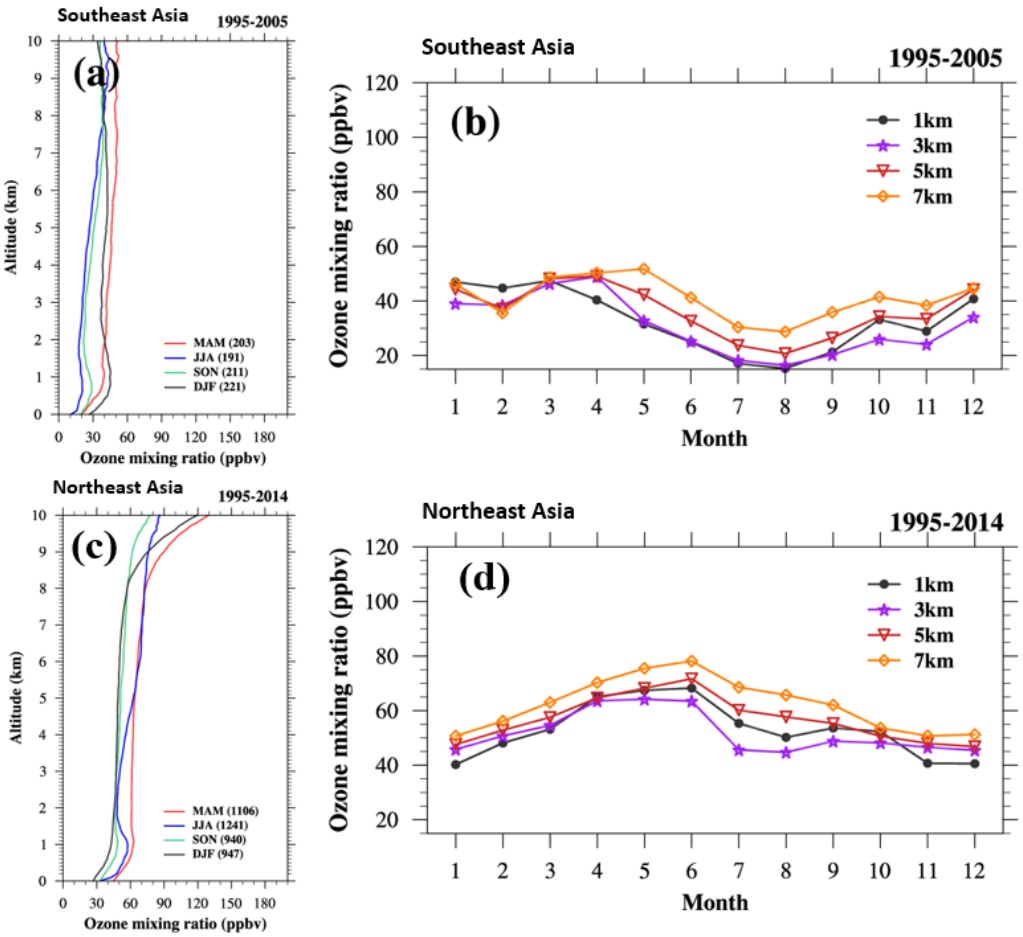

**Figure 13**. Analysis of the IAGOS measurements: (a) Seasonal mean vertical ozone profiles in Southeast Asia during March-April-May (MAM, red), June-July-August (JJA, blue), September-October-November (SON, green), and December-January-February (DJF, black), (b) monthly mean ozone variation of 1-km (black), 3-km (purple), 5-km (red), and 7-km (orange) altitudes in Southeast Asia, (c) seasonal mean vertical ozone profiles in Northeast Asia during MAM (red), JJA (blue), SON (green), and DJF (black), and (d) Monthly mean ozone variation of 1-km (black), 3-km (purple), 5-km (red), and 7-km (orange) altitudes in Northeast Asia.



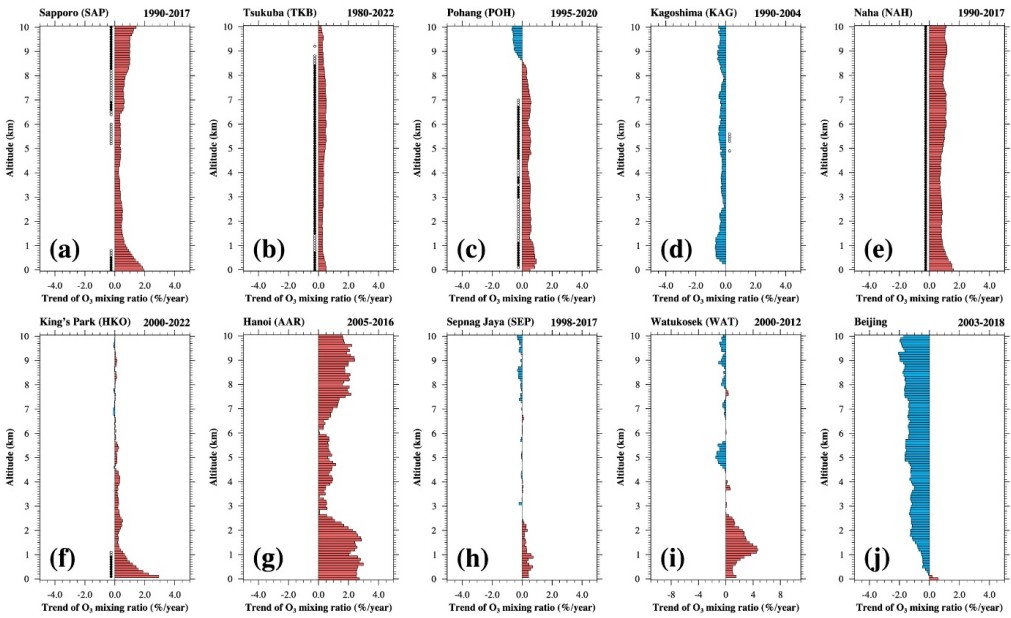

**Figure 14**. Long-term trends of annual mean ozone per 100-m range from 0 to 10 km altitude at (a) Sapporo (SAP), (b) Tsukuba (TKB), (c) Pohang (POH), (d) Kagoshima (KAG), (e) Naha (NAH), (f) King's park (HKO), (g) Hanoi (AAR), (h) Sepang Jaya (SEP), (i) Watukosek, and (j) Beijing site. Orange color means increasing, and blue color means decreasing trend. Black dot indicates that the trend is statistically significant having a *p*-value smaller than 0.01, and white dot does that the trend is statistically significant having a *p*-value between 0.01 and 0.05.



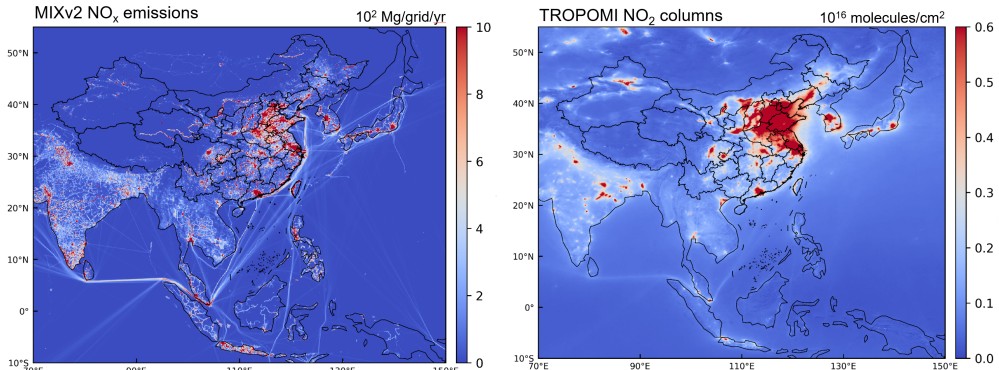

**Figure 15.** The spatial distribution of bottom-up NO$_x$ emissions from MIXv2 inventory (left) and the TROPOMI satellite derived NO$_2$ columns (right). Due to the data availability, emission data for year 2017 and satellite data for year 2019 are used to represent the present-day level (2017-2021), respectively.



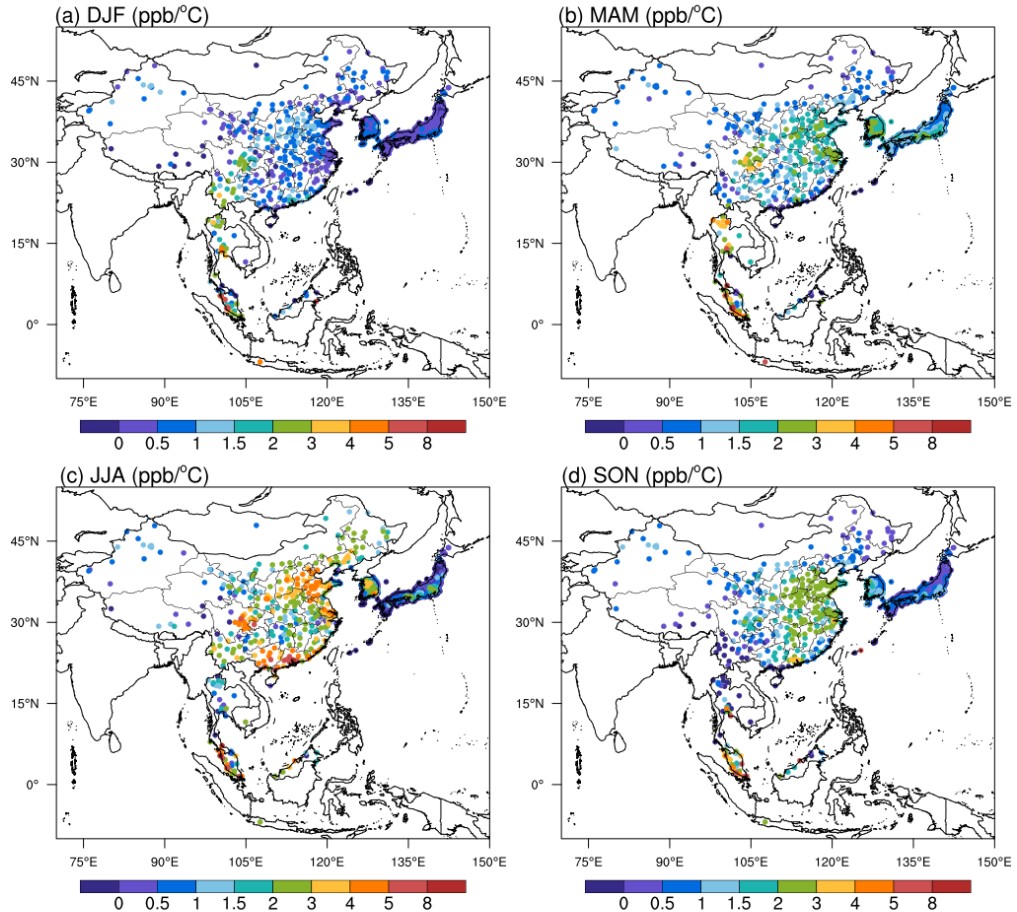

**Figure 16.** The observed 50th percentile regression slope (ppb °C$^{-1}$) between daily surface MDA8 ozone and daily maximum 2-m air temperature in (a) DJF, (b) MAM, (c) JJA, and (d) SON averaged over 2017-2021.