# Peer review of "Surface and tropospheric ozone over East Asia and Southeast Asia from"

_EGUsphere, 2024_

## Referee Comment (RC2)

Review of Li et al., 2025

In this paper, the authors discuss ozone measurements and trends over East and Southeast Asia. This data was collected as part of the Tropospheric Ozone Assessment Report (TOAR) project and includes surface measurements and vertical profiles of ozone. The authors compute ozone distribution, trends, and exceedances for many regions of Asia which were previously under-sampled. The authors also highlight the role of stratospheric ozone intrusions and the reality of long-term ozone exposure over much of East and Southeast Asia. I believe this paper is scientifically sound and presents actionable results for air quality regulation in the participating countries.

Therefore, I recommend that this paper be accepted with the following **minor** revisions.

Specific Comments:

1) The introduction is very comprehensive about why we should care about tropospheric ozone, and it does a good job summarizing ozone trends over Asia. However, I would like the authors to go into further details about *why* the TOAR database is important.  Does it fill data gaps in space and/or time? Is it a convenient new dataset available to the community? This paper was my first exposure to the TOAR project, and I was still left with some of these basic questions. Especially since the intro does such a thorough job explaining ozone trends and the paper continues to split ozone metrics/trends by country, it is unclear to me what value the TOAR dataset adds. Could you comment in the conclusions/discussion as well on what future work could do with the TOAR data? Some ideas which come to my mind that TOAR could be useful for are to analyze ozone trend by lat/lon bins, rural/urban bins, and coastal/inland bins, where political boundaries are less important and some physics questions can be answered.

2) Section 3.1.2: How "new" is the WHO peak season ozone trend? Are there previous studies you can compare to or does the TOAR data allow this to be calculated in a unique way. If this is brand new, highlight this very useful finding!

3) I really like the spatial maps of seasonal ozone concentrations and exceedances. These are very clear and well-explained. At the same time, the paper could benefit from some figures being reorganized or removed. Please see below.

   a. Figure 10: I would recommend moving this to Figure 1 since it is the first figure referenced in the flow of the paper

   b. Figure 3: is this figure necessary? The ozone standards are already mentioned in section 2 and in the supplement S2. I can also see how it be helpful to mention this figure or Figure S2 in section 2 if the authors want to keep this information in the paper visually.

   c. Figure 6 (and other trend figures): There is a lot of information contained in these figures, and it took me a bit to get a handle on what was being shown. I think the following might help to make these figures more digestible: 1) small + and – signs added above the colors, to indicate that blue is decreasing and

red is increasing. 2) increase the size of the arrow legend showing the trend per year. 3) make colored arrows smaller/thinner. These arrows often sit on top of each other and obscure regional variability.

4) Overall, please refer to specific figure panels when appropriate.
5) I would recommend a different title for section 4 that makes it clearer that the paper will be discussing vertical profiles.
6) Lines 428-442: The idea of the ozone climate penalty seems to me to be more in line with "current ozone distributions" or "ozone trends". Maybe consider highlighting the climate penalty earlier in the paper.

Technical Corrections:
Line 146: change "8h average" to "1h average" since you are computing MDA1
Line 160: replace "continued" with "operational" or "ongoing"
Line 161: remove "for utilizing"
Line 174: "regress" should be "regression"
Line 319: change "Figure 10" to "Figure 1" if the figure gets moved
Line 327: change "is used to be strongly intruded" to "is strongly intruding"
Line 398: remove "In a same way"
Line 407: replace "whole" with "to an overall"
Line 411: delete "At surface,"
Line 433: "slop" should be "slope"

---

## Author Comment (AC1)

**Response to Referee #2**

Review of Li et al., 2025

In this paper, the authors discuss ozone measurements and trends over East and Southeast Asia. This data was collected as part of the Tropospheric Ozone Assessment Report (TOAR) project and includes surface measurements and vertical profiles of ozone. The authors compute ozone distribution, trends, and exceedances for many regions of Asia which were previously under-sampled. The authors also highlight the role of stratospheric ozone intrusions and the reality of long-term ozone exposure over much of East and Southeast Asia. I believe this paper is scientifically sound and presents actionable results for air quality regulation in the participating countries.

Therefore, I recommend that this paper be accepted with the following **minor** revisions.

Thank you very much for your very constructive comments. We have addressed them carefully and please find our point-by-point response in blue.

Specific Comments:

1) The introduction is very comprehensive about why we should care about tropospheric ozone, and it does a good job summarizing ozone trends over Asia. However, I would like the authors to go into further details about why the TOAR database is important. Does it fill data gaps in space and/or time? Is it a convenient new dataset available to the community? This paper was my first exposure to the TOAR project, and I was still left with some of these basic questions. Especially since the intro does such a thorough job explaining ozone trends and the paper continues to split ozone metrics/trends by country, it is unclear to me what value the TOAR dataset adds. Could you comment in the conclusions/discussion as well on what future work could do with the TOAR data? Some ideas which come to my mind that TOAR could be useful for are to analyze ozone trend by lat/lon bins, rural/urban bins, and coastal/inland bins, where political boundaries are less important and some physics questions can be answered.

The TOAR (https://igacproject.org/activities/TOAR/) was created by the IGAC on 2014 and it's goals are to assess tropospheric ozone from available measurements and constructed a freely accessible ozone database. The TOAR provides the global largest database for surface ozone records and we can examine global ozone levels within the same time frame.

We have added some description about the importance of TOAR in Lines 127-132: "The TOAR data portal archives a global comprehensive and freely accessible data collection of surface ozone observations (https://igacproject.org/activities/TOAR/TOAR-II), which supports TOAR's assessment report of global ozone distributions and trends from surface to the tropopause. The TOAR database keeps updated to include all recent observations since 2014. To give an up-to-date assessment of tropospheric ozone over East Asia and Southeast Asia, here we take advantage of TOAR database to examine ozone levels in different countries within the same time frame".

Thanks for your great suggestion for future work. In fact, more analysis using TOAR data have been or is being conducted by other researcher. For example, more papers from Tropospheric

Ozone Assessment Report Phase II (TOAR-II) Community Special Issue (https://amt.copernicus.org/articles/special_issue10_1256.html). And key TOAR-II Assessment papers on global ozone levels with focused analysis (e.g., lat/lon bins) are being developed by TOAR II team (https://igacproject.org/activities/TOAR/TOAR-II). As such, to avoid confusion, we decide not to comment on future studies in our manuscript.

2) Section 3.1.2: How "new" is the WHO peak season ozone trend? Are there previous studies you can compare to or does the TOAR data allow this to be calculated in a unique way. If this is brand new, highlight this very useful finding!

Thanks for noting this! The WHO peak season ozone standard was introduced in September 2021 and no previous studies assessed this metric in Asia as far as we know.

We highlighted this in Lines 258-260: "we apply the new WHO standard for peak season ozone that was introduced in September 2021 to assess risks of long-term ozone exposure over East Asia and Southeast Asia, which has not been examined in previous studies."

3) I really like the spatial maps of seasonal ozone concentrations and exceedances. These are very clear and well-explained. At the same time, the paper could benefit from some figures being reorganized or removed. Please see below.

  a. Figure 10: I would recommend moving this to Figure 1 since it is the first figure referenced in the flow of the paper

  Thanks for the suggestion. Figure 10 only provides information for ozone profile measurements and we would like to introduce it after the surface ozone part. To avoid confusion, we revised the text in Line 175: "Location of all ozonesonde sites and the IAGOS region will be detailed in Section 4.1"

  b. Figure 3: is this figure necessary? The ozone standards are already mentioned in section 2 and in the supplement S2. I can also see how it be helpful to mention this figure or Figure S2 in section 2 if the authors want to keep this information in the paper visually.

  Yes, we want to keep it to demonstrate this diverse ozone AQ standard. In particular, we want to highlight that in Lines 495-496: "The diverse short-term ozone air quality standards in Southeast Asian countries (Figure 3) suggest a great challenge to call for regional joint ozone control."

  c. Figure 6 (and other trend figures): There is a lot of information contained in these figures, and it took me a bit to get a handle on what was being shown. I think the following might help to make these figures more digestible: 1) small + and – signs added above the colors, to indicate that blue is decreasing and red is increasing. 2) increase the size of the arrow legend showing the trend per year.

  Well taken! The trend plots have been updated as suggested!

  3) make colored arrows smaller/thinner. These arrows often sit on top of each other and obscure regional variability.

  We have made them thinner but there are still some overlaps due to regional dense stations.

4) Overall, please refer to specific figure panels when appropriate.

Added.

5) I would recommend a different title for section 4 that makes it clearer that the paper will be discussing vertical profiles.

We have changed it to: "4. Present-day distribution and long-term trends in tropospheric ozone profiles"

6) Lines 428-442: The idea of the ozone climate penalty seems to me to be more in line with "current ozone distributions" or "ozone trends". Maybe consider highlighting the climate penalty earlier in the paper.

Agreed. Now we have moved this part in Section 3. Please find it in Lines 278-294.

Technical Corrections:

Line 146: change "8h average" to "1h average" since you are computing MDA1

Line 160: replace "continued" with "operational" or "ongoing"

Line 161: remove "for utilizing"

Line 174: "regress" should be "regression"

Line 319: change "Figure 10" to "Figure 1" if the figure gets moved

Line 327: change "is used to be strongly intruded" to "is strongly intruding"

Line 398: remove "In a same way"

Line 407: replace "whole" with "to an overall"

Line 411: delete "At surface,"

Line 433: "slop" should be "slope"

Many thanks and we have walked through these grammar corrections.

---

## Author Comment (AC2)

**Response to Referee #1**

This is an excellent paper about the ozone concentrations in East and Southeast Asia. Key features are the quantity of surface stations in eight countries with different air quality standards, together with ozonesonde data from 10 sites. Another noticeable feature is the period investigated, which extended above 10 years in some sites. The authors presented the annual cycle of mean values, the 95[th] percentile, the number of threshold exceedances and trends. Tropospheric ozone profiles are presented and their trends. Finally, a relationship with nitrogen oxides and temperature is considered. Since the paper is quite complete, only some minor comments should be required before its final acceptance.

Thank you very much for your very helpful comments. We have addressed them carefully and please find our point-by-point response in blue.

Since the number of surface stations included in this study is noticeable, the authors could comment if noticeable outliers have been recorded, i.e. if there are stations that provide anomalous values.

Thanks for your comments.

For surface ozone, we have clarified that noticeable outliers are not detected in our dataset after data quality control. We show the observed maximum daily MDA8 ozone during 2017-2021 in different seasons in Figure R1, and find their ratios to seasonal median ozone are almost within 3~4, which suggests unnoticeable outliers for surface ozone.

For ozonesonde and IAGOS dataset, noticeable outliers are also not clearly detected. Instead, the data sampling shows large annual difference, which can be associated with the interpretation of our results. Thus, in this revision process, we have added the number of all ozonesonde and IAGOS data used in this study in Figure S1.

We have also added this information in Lines 179-180:

"Noticeable outliers are not detected in our dataset for both surface ozone and ozonesonde and IAGOS datasets."

[Figure]

Figure R1. The ratios of the observed maximum daily MDA8 ozone during 2017-2021 to seasonal median ozone in (a) DJF, (b) MAM, (c) JJA, and (d) SON over East Asia and Southeast Asia.

[Figure]

Figure S1. The number of all (a) ozonesonde and (b) IAGOS data used in this study.

The paper focus is on the high values. However, low values could be highlighted. The authors should indicate if such values belong to remote sites or if are linked to high concentrations of other substances.

Thanks for the suggestion! We have added some description about the low ozone values.

For surface ozone, in Lines 199-201: "In many Chinese cities, ozone concentration is even decreased to 20-30 nmol mol$^{-1}$, and this is because the high NO$_x$ emissions in urban environment (e.g., North China Plain) make ozone strongly titrated",

Lines 231-233: "However, Borneo Malaysia and Indonesia still record the 95th percentile ozone lower than 50 nmol mol$^{-1}$, suggesting the important role of fresh marine air inflow."

Lines 283-286: "In contrast, low ozone-temperature slope of less than 1 nmol mol$^{-1}$ $°C^{-1}$ across different seasons can be also found in some sites over Japan and Tibetan Plateau of China, suggesting a minimal role of local ozone photochemical formation in these remote sites."

For tropospheric ozone, in Lines 413-416: "However, ozone in winter (DJF) is not the lowest but ozone in summer (JJA) is the lowest in Southeast Asia, probably due to the relatively stronger precipitation in summer, and warmer temperature in winter, compared to the atmospheric condition in Northeast Asia."

The authors could indicate if it is possible to classify the stations following the ozone origin by transport or by precursors or if site classifications have been discarded.

This is a very important point classifying the stations by their ozone sources, although it is beyond of the scope this paper.

We have added in Lines 303-305: "It also deserves further study of cluster analysis about the ozone origin by transport or by precursors by taking advantage of this considerable ozone data records".

Since the number of stations depends on the country. Some means presented in Figure 8 may be more robust against others. The authors could consider this fact.

Thanks! Following your suggestion, we have added country-level ozone means in Figure 8 and Figure 6.

Finally, the temperature is considered. However, meteorological features are varied in different latitudes. The synoptic pattern evolution may be quite different in the analysed region. The authors could comment the possible influence of such features on ozone concentrations.

Following the referee's suggestion, we have conducted additional analysis on the relationship between ozone and other meteorological features. The added Figure Sx-Sx are the calculated correlations between ozone and other key meteorological variables. Some related description has been also added in Lines 295-303:

"Considering the meteorological features may be quite different in different latitudes, we conducted additional analysis on the relationship between ozone and other meteorological features (Figure S3-S7). The widespread positive (negative) correlation between ozone and temperature (relative humidity), reflecting the known conductive condition for ozone photochemistry. However, the synoptic patterns that are important for ozone transport varied greatly at a regional scale. For example, in Figure S6, summertime southerly winds are conducive for ozone pollution over North China by transporting ozone precursors and warmer

air, but would decrease ozone over Southern China by carrying with cleaner marine inflow. As such, identifying the key synoptic pattern will be also necessary for understanding local ozone variations under climate change."

[Figure]

**Figure S3.** The correlation coefficients between observed daily surface MDA8 ozone and daily maximum 2-m air temperature in (a) DJF, (b) MAM, (c) JJA, and (d) SON averaged over 2017-2021.

[Figure]

**Figure S4.** Same with Figure S3, but for relative humidity (RH)

[Figure]

**Figure S5.** Same with Figure S3, but for 10-m zonal wind (U10).

[Figure]

**Figure S6.** Same with Figure Ss, but for 10-m meridional wind (V10).

[Figure]

**Figure S7.** Same with Figure S3, but for sea level pressure (SLP)

Minor remarks.

L. 403. Dot instead of semicolon.

Corrected.

L. 580. Introduce one space in "forthe".

Done.

L. 583. Introduce subscript in $NO_2$.

Done.

---

## Author Comment (AC3)

**Response to community comments by Owen Copper**

Comments by Owen R. Cooper (TOAR Scientific Coordinator of the Community Special Issue) on:

**Surface and tropospheric ozone over East Asia and Southeast Asia from observations: distributions, trends, and variability**

Ke Li, Rong Tan, Wenhao Qiao, Taegyung Lee, Yufen Wang, Danyuting Zhang, Minglong Tang, Wenqing Zhao, Yixuan Gu, Shaojia Fan, Jinqiang Zhang, Xiaopu Lyu, Likun Xue, Jianming Xu, Zhiqiang Ma, Mohd Talib Latif, Teerachai Amnuaylojaroen, Junsu Gil, Mee-Hye Lee, Juseon Bak, Joowan Kim, Hong Liao, Yugo Kanaya, Xiao Lu, Tatsuya Nagashima, and Ja-Ho Koo

EGUsphere [preprint], https://doi.org/10.5194/egusphere-2024-3756

Discussion started: 21 Jan 2025

Discussion closes: 04 Mar 2025

This review is by Owen Cooper, TOAR Scientific Coordinator of the TOAR-II Community Special Issue. I, or a member of the TOAR-II Steering Committee, will post comments on all papers submitted to the TOAR-II Community Special Issue, which is an inter-journal special issue accommodating submissions to six Copernicus journals: ACP (lead journal), AMT, GMD, ESSD, ASCMO and BG. The primary purpose of these reviews is to identify any discrepancies across the TOAR-II submissions, and to allow the author teams time to address the discrepancies. Additional comments may be included with the reviews. While O. Cooper and members of the TOAR Steering Committee may post open comments on papers submitted to the TOAR-II Community Special Issue, they are not involved with the decision to accept or reject a paper for publication, which is entirely handled by the journal's editorial team.

Thanks for your attention so much. We sincerely prepared our responses about your comments, and suggested those as below.

**Comments regarding TOAR-II guidelines:**

TOAR-II has produced two guidance documents to help authors develop their manuscripts so that results can be consistently compared across the wide range of studies that will be written for the TOARII Community Special Issue. Both guidance documents can be found on the TOAR-II webpage: (https://igacproject.org/activities/TOAR/TOAR-II).

The TOAR-II Community Special Issue Guidelines: In the spirit of collaboration and to allow TOAR-II findings to be directly comparable across publications, the TOAR-II Steering Committee has issued this set of guidelines regarding style, units, plotting scales, regional and tropospheric column comparisons, and tropopause definitions.

The TOAR-II Recommendations for Statistical Analyses: The aim of this guidance note is to provide recommendations on best statistical practices and to ensure consistent communication of statistical analysis and associated uncertainty across TOAR publications. The scope includes

approaches for reporting trends, a discussion of strengths and weaknesses of commonly used techniques, and calibrated language for the communication of uncertainty. Table 3 of the TOAR-II statistical guidelines provides calibrated language for describing trends and uncertainty, similar to the approach of IPCC, which allows trends to be discussed without having to use the problematic expression, "statistically significant".

Thanks for your helpful information so much. We tried our best for following this guideline. In the previous manuscript, generally our analyses are in the range of TOAR-II guidance but some of our analyses were not well matching to the guideline style and the recommended way of statistical analyses.

In this revision process, we now have improved our manuscript based on these two guidance documents. Main improvements are as follows:

1) Recommended color-scale is considered for the Figures. 2) We have the unit of ozone concentrations corrected to the nmol mol$^{-1}$ except for the national ozone air quality standard. 3) Quantile regression methods are adopted for all of the trend analysis. 4) More calibrated language is used now by avoiding the statement as "statistically significance".

**General comments:**

Line 154: Here the authors state that the ozonesonde stations have at least 10 profiles per year, which does not tell us the typical sampling rate for each station. As shown in a recent paper published in the TOAR-II Community Special Issue (Chang et al., 2024), accurate detection of ozone trends in the free troposphere is not possible if sample sizes are very low. This issue was further addressed by Gaudel et al. (2024) (another TOAR-II paper) who provided confidence levels for the ozone trends calculated from ozonesondes and IAGOS data across the tropics (see their Table 1); they also provided many details regarding the sampling frequency of each time series. It would be helpful to the reader if the approach of Chang et al. (2024) and Gaudel et al. (2024) is taken into consideration by the authors of the submitted paper. Please provide greater details regarding the sampling frequency at each station, and also provide some discussion regarding your confidence in the reported trends.

Thanks for your comments. Now we added the information about the sample sizes for the ozonesonde and IAGOS data. This information is now included in the updated supplements based on the bar-plot format due to the better visualization in Figure S1. Here in the response document, we also added the number information as below (Figure R1 for ozonesonde data and Figures R2-R3 for IAGOS data).

| Year/Site | Sapporo | Tsukuba | Pohang | Kagoshima | Naha | Taipei | King's Park | Hanoi | Kuala Lumpur | Watukosek | Beijing |
|---|---|---|---|---|---|---|---|---|---|---|---|
| 1980 | 0 | 16 | 0 | 0 | 0 | 0 | 0 | 0 | 0 | 0 | 0 |
| 1981 | 0 | 17 | 0 | 0 | 0 | 0 | 0 | 0 | 0 | 0 | 0 |
| 1982 | 0 | 14 | 0 | 0 | 0 | 0 | 0 | 0 | 0 | 0 | 0 |
| 1983 | 0 | 18 | 0 | 0 | 0 | 0 | 0 | 0 | 0 | 0 | 0 |
| 1984 | 0 | 16 | 0 | 0 | 0 | 0 | 0 | 0 | 0 | 0 | 0 |
| 1985 | 0 | 15 | 0 | 0 | 0 | 0 | 0 | 0 | 0 | 0 | 0 |
| 1986 | 0 | 15 | 0 | 0 | 0 | 0 | 0 | 0 | 0 | 0 | 0 |
| 1987 | 0 | 17 | 0 | 0 | 0 | 0 | 0 | 0 | 0 | 0 | 0 |
| 1988 | 0 | 22 | 0 | 0 | 0 | 0 | 0 | 0 | 0 | 0 | 0 |
| 1989 | 0 | 24 | 0 | 0 | 0 | 0 | 0 | 0 | 0 | 0 | 0 |
| 1990 | 15 | 35 | 0 | 13 | 15 | 0 | 0 | 0 | 0 | 0 | 0 |
| 1991 | 23 | 48 | 0 | 26 | 43 | 0 | 0 | 0 | 0 | 0 | 0 |
| 1992 | 34 | 56 | 0 | 25 | 31 | 0 | 0 | 0 | 0 | 0 | 0 |
| 1993 | 42 | 52 | 0 | 36 | 41 | 0 | 0 | 0 | 0 | 0 | 0 |
| 1994 | 45 | 49 | 0 | 35 | 40 | 0 | 0 | 0 | 0 | 0 | 0 |
| 1995 | 47 | 57 | 40 | 41 | 37 | 0 | 0 | 0 | 0 | 0 | 0 |
| 1996 | 38 | 57 | 15 | 43 | 36 | 0 | 0 | 0 | 0 | 0 | 0 |
| 1997 | 43 | 70 | 42 | 35 | 40 | 0 | 0 | 0 | 0 | 0 | 0 |
| 1998 | 45 | 61 | 44 | 41 | 41 | 0 | 0 | 0 | 21 | 22 | 0 |
| 1999 | 43 | 54 | 43 | 47 | 42 | 0 | 0 | 0 | 21 | 21 | 0 |
| 2000 | 45 | 54 | 37 | 50 | 41 | 31 | 45 | 0 | 25 | 44 | 0 |
| 2001 | 46 | 73 | 41 | 46 | 41 | 33 | 14 | 0 | 27 | 41 | 45 |
| 2002 | 49 | 66 | 34 | 52 | 43 | 0 | 12 | 0 | 27 | 21 | 32 |
| 2003 | 48 | 46 | 30 | 46 | 44 | 0 | 41 | 0 | 24 | 44 | 54 |
| 2004 | 44 | 51 | 44 | 48 | 39 | 0 | 62 | 11 | 24 | 39 | 59 |
| 2005 | 47 | 47 | 48 | 10 | 41 | 0 | 51 | 12 | 21 | 7 | 69 |
| 2006 | 45 | 44 | 44 | 0 | 43 | 0 | 43 | 37 | 21 | 8 | 53 |
| 2007 | 38 | 44 | 46 | 0 | 38 | 0 | 44 | 28 | 23 | 15 | 45 |
| 2008 | 42 | 49 | 51 | 0 | 37 | 0 | 49 | 23 | 24 | 24 | 37 |
| 2009 | 34 | 48 | 45 | 0 | 40 | 0 | 50 | 21 | 23 | 15 | 48 |
| 2010 | 39 | 48 | 43 | 0 | 37 | 0 | 47 | 10 | 4 | 10 | 48 |
| 2011 | 45 | 50 | 46 | 0 | 43 | 0 | 51 | 11 | 0 | 11 | 47 |
| 2012 | 49 | 49 | 36 | 0 | 48 | 0 | 49 | 8 | 16 | 12 | 41 |
| 2013 | 44 | 45 | 33 | 0 | 49 | 0 | 47 | 20 | 19 | 9 | 59 |
| 2014 | 41 | 43 | 42 | 0 | 36 | 16 | 50 | 21 | 24 | 0 | 45 |
| 2015 | 51 | 48 | 40 | 0 | 48 | 12 | 48 | 22 | 19 | 0 | 43 |
| 2016 | 49 | 47 | 37 | 0 | 46 | 12 | 48 | 21 | 24 | 0 | 47 |
| 2017 | 47 | 43 | 35 | 0 | 48 | 14 | 37 | 11 | 24 | 0 | 46 |
| 2018 | 5 | 33 | 40 | 0 | 5 | 7 | 45 | 30 | 20 | 0 | 46 |
| 2019 | 0 | 43 | 45 | 0 | 0 | 6 | 48 | 23 | 12 | 0 | 5 |
| 2020 | 0 | 40 | 48 | 0 | 0 | 6 | 48 | 28 | 13 | 0 | 0 |
| 2021 | 0 | 42 | 0 | 0 | 0 | 7 | 48 | 13 | 21 | 14 | 0 |
| 2022 | 0 | 39 | 0 | 0 | 0 | 3 | 46 | 0 | 24 | 13 | 0 |

Figure R1. The number of ozonesonde data used for the analysis in this research.

In previous version, we did not consider the year having ozonesonde data less than 10, and also did not conduct the trend analysis for a site not having at least 5 continual years with data number higher than 10. Using this criteria, we intended to make better reliability of our results. But we recognized that this approach makes somewhat large restriction of data usage. Considering that the ozonesonde data is still rare and difficult to obtain, we now agree with the commnets and think it is much better to use all measured data as possible as we can.

Now, we actually repeated all analyses in a same way, only having different dataset and compared previous and updated results (data used based on our previous standard vs. all data used). As a result, we confirmed that whole analyzed results about the ozonesonde data are not different much – Main lessons are almost identical, just the p-value part changes a little.

In conclusion, we updated all result figures (main manuscript and supplements) using all ozonesonde data (i.e., including a year if data number < 10). With proper citation of recommended results, we have revised our statements in Lines 161-170:

"There are a number of ozonesonde measurement sites, but here, we only consider 10 sites (Table S3), which has at least 10 measurement years continuously for finding reliable trends by considering the lesson from Chang et al. (2024). If a certain site is not satisfying with this standard, we only suggest the mean pattern of ozone vertical profile for that site in order to show all existed data as possible as we can. This approach enables us to compare with recent results

produced from other ozonesonde data analyses (Gaudel et al., 2024; Stauffer et al., 2024). Data at 9 sites were obtained from the World Ozone and Ultraviolet Radiation Data Centre (WOUDC) and Southern Hemisphere ADditional OZonesondes (SHADOZ) data archive. Data at Beijing site, which were reported in the previous study (Zhang et al., 2021) were directly provided from the measurement team."

Please check the data availability for the ozonesonde sites of Hanoi, Sepang Jaya and Watukosek, as the information provided in Figure S2 indicates that these data sets are incomplete. These sites are part of the NASA SHADOZ ozonesonde network, and the primary database for these data is here: https://tropo.gsfc.nasa.gov/shadoz/Archive.html

According to the SHADOZ database, these sites have data through 2021 or 2022 (see Stauffer et al. 2024 in the TOAR-II Community Special Issue). For your study please download and use the complete time series. Please also follow the SHADOZ data use guidelines at the bottom of the URL listed above.

Thanks! Previously we only checked the WOUDC data archive, and we now recognized that some SHADOZ data were not successfully transferred to the WOUDC archive (We discussed about this issue with Dr. Ryan Stauffer, who is one of expert scientists in the SHADOZ group).

Now we have used the complete dataset of ozonesonde measurements and updated our results. The key things related to this updating:

1) Site name change: Sepang Jaya => Kuala Lumpur.

2) The period of used data in some sites: Hanoi (2005-2016 => 2005-2021), Watukosek (2000-2012 => 1998-2022), and Kuala Lumpur (1998-2017 => 1998-2022).

3) Trend patterns: All trends are provided with their p-value. For Hanoi, increasing trend is same, but the trend slope becomes smaller. For Kuala Lumpur and Watukosek, trend patterns are almost similar to previous results

In addition, acknowledgements are also added into the text in according to the SHADOZ data use guidelines.

IAGOS data: The authors state that there are very few IAGOS profiles available over Asia after 2014, however there are actually hundreds of available profiles over East Asia and Southeast Asia, as shown in the recent TOAR-II submission by Lu et al., 2024. For example, I went to the IAGOS database and downloaded all of the ozone and carbon monoxide profiles above northeast China for the period 2015- 2022 (this region does not include South Korea or Japan). I found a total of 526 profiles, as shown in the figure below. This is equal to 66 profiles per year, which is a greater sampling frequency than the typical ozonesonde station that launches once per week. Please include the 2015-2022 IAGOS data in your analysis. Please also ensure that the IAGOS data policy has been followed, which is copied below.

Thanks for helpful comments! When we first performed the IAGOS analysis, our concern is the large difference of data number among different years: enough data in some years, but not in other years. Two figures attached below show the number of IAGOS sampling for Northeast and

Southeast Asia, respectively (Figures R2 and R3). This information is newly added to the supplement as Figure S1.

Since the sampling number does not show the inter-annual consistence, in our initial analysis we only considered the year having > 100 IAGOS measurements. That is the reason that our submitted results have short time period (also there was a typo, not 2014 but 2018).

In this revision, however, we follow your comment to use all available data. As a result, we can now extend the trend analysis until the year 2022. This enables us to do the trend analysis in the Southeastern Asia region, too! Main lessons that we found does not change, but it looks much better to show results based on completed data. The text in Lines 398-421 have been updated.

| | 1 | 2 | 3 | 4 | 5 | 6 | 7 | 8 | 9 | 10 | 11 | 12 | Sum |
|---|---|---|---|---|---|---|---|---|---|---|---|---|---|
| 1995 | 2 | 3 | 7 | 16 | 21 | 15 | 28 | 16 | 26 | 17 | 17 | 19 | 187 |
| 1996 | 26 | 24 | 13 | 35 | 30 | 28 | 31 | 37 | 22 | 28 | 19 | 23 | 316 |
| 1997 | 37 | 33 | 33 | 31 | 37 | 48 | 24 | 51 | 22 | 45 | 49 | 58 | 468 |
| 1998 | 55 | 47 | 51 | 43 | 68 | 51 | 54 | 67 | 33 | 48 | 51 | 52 | 620 |
| 1999 | 41 | 54 | 22 | 34 | 28 | 24 | 39 | 29 | 33 | 35 | 19 | 20 | 378 |
| 2000 | 16 | 18 | 13 | 21 | 15 | 33 | 34 | 28 | 23 | 21 | 21 | 23 | 266 |
| 2001 | 10 | 15 | 23 | 20 | 16 | 13 | 11 | 8 | 9 | 15 | 17 | 2 | 159 |
| 2002 | 11 | 6 | 2 | 9 | 8 | 3 | 22 | 24 | 22 | 21 | 17 | 9 | 154 |
| 2003 | 18 | 22 | 25 | 22 | 26 | 27 | 24 | 22 | 20 | 18 | 20 | 21 | 265 |
| 2004 | 19 | 15 | 15 | 15 | 16 | 20 | 25 | 25 | 2 | 7 | 19 | 24 | 202 |
| 2005 | 11 | 9 | 40 | 51 | 44 | 53 | 41 | 38 | 33 | 38 | 11 | 9 | 378 |
| 2006 | 9 | 1 | 1 | 1 | 7 | 12 | 9 | 11 | 9 | 9 | 0 | 0 | 69 |
| 2007 | 0 | 0 | 0 | 0 | 0 | 0 | 0 | 1 | 0 | 1 | 0 | 0 | 2 |
| 2008 | 0 | 0 | 0 | 0 | 0 | 0 | 0 | 2 | 0 | 2 | 0 | 2 | 6 |
| 2009 | 1 | 2 | 2 | 0 | 2 | 0 | 0 | 0 | 0 | 0 | 4 | 8 | 19 |
| 2010 | 3 | 5 | 0 | 0 | 0 | 2 | 2 | 1 | 1 | 2 | 11 | 6 | 33 |
| 2011 | 12 | 13 | 22 | 25 | 11 | 20 | 20 | 15 | 14 | 17 | 13 | 8 | 190 |
| 2012 | 0 | 3 | 9 | 37 | 15 | 0 | 36 | 50 | 54 | 14 | 16 | 20 | 254 |
| 2013 | 29 | 12 | 2 | 20 | 39 | 25 | 30 | 22 | 6 | 6 | 8 | 42 | 241 |
| 2014 | 23 | 23 | 30 | 22 | 19 | 28 | 9 | 18 | 6 | 4 | 0 | 5 | 187 |
| 2015 | 17 | 14 | 22 | 17 | 0 | 5 | 6 | 19 | 22 | 28 | 14 | 29 | 193 |
| 2016 | 29 | 9 | 12 | 14 | 9 | 16 | 63 | 49 | 59 | 77 | 53 | 53 | 443 |
| 2017 | 19 | 15 | 0 | 10 | 14 | 10 | 12 | 19 | 11 | 6 | 15 | 70 | 201 |
| 2018 | 59 | 39 | 37 | 37 | 39 | 26 | 20 | 21 | 10 | 18 | 0 | 10 | 316 |
| 2019 | 12 | 16 | 6 | 1 | 7 | 8 | 10 | 4 | 0 | 0 | 2 | 9 | 75 |
| 2020 | 5 | 6 | 0 | 0 | 30 | 1 | 11 | 6 | 0 | 0 | 0 | 1 | 60 |
| 2021 | 2 | 1 | 0 | 3 | 8 | 0 | 0 | 6 | 0 | 0 | 0 | 0 | 20 |
| 2022 | 2 | 1 | 4 | 0 | 0 | 0 | 0 | 0 | 2 | 2 | 2 | 0 | 13 |

Figure R2. The number of IAGOS data sampled in the defined region as Northeast Asia.

|      | 1 | 2 | 3 | 4 | 5 | 6 | 7 | 8 | 9 | 10 | 11 | 12 | Sum |
|------|---|---|---|---|---|---|---|---|---|----|----|----|-----|
| 1995 | 4 | 16 | 18 | 6 | 4 | 8 | 24 | 14 | 35 | 3 | 11 | 9 | 152 |
| 1996 | 31 | 33 | 3 | 25 | 28 | 24 | 23 | 16 | 0 | 32 | 24 | 16 | 255 |
| 1997 | 35 | 36 | 44 | 4 | 2 | 10 | 10 | 6 | 6 | 1 | 3 | 8 | 165 |
| 1998 | 12 | 4 | 6 | 22 | 6 | 4 | 8 | 8 | 2 | 10 | 2 | 12 | 96 |
| 1999 | 5 | 7 | 14 | 0 | 0 | 12 | 7 | 2 | 4 | 6 | 2 | 4 | 63 |
| 2000 | 0 | 2 | 8 | 2 | 2 | 0 | 2 | 2 | 0 | 1 | 0 | 3 | 22 |
| 2001 | 0 | 0 | 0 | 0 | 0 | 0 | 0 | 0 | 0 | 0 | 0 | 0 | 0 |
| 2002 | 2 | 0 | 2 | 0 | 0 | 0 | 0 | 4 | 0 | 0 | 0 | 2 | 10 |
| 2003 | 0 | 0 | 2 | 0 | 0 | 0 | 0 | 4 | 4 | 0 | 0 | 0 | 10 |
| 2004 | 0 | 2 | 0 | 0 | 0 | 4 | 2 | 1 | 1 | 4 | 7 | 3 | 24 |
| 2005 | 13 | 6 | 9 | 15 | 45 | 40 | 6 | 10 | 46 | 38 | 16 | 14 | 258 |
| 2006 | 6 | 0 | 0 | 4 | 13 | 16 | 23 | 21 | 12 | 15 | 4 | 4 | 118 |
| 2007 | 0 | 4 | 4 | 4 | 4 | 3 | 2 | 4 | 0 | 4 | 3 | 0 | 32 |
| 2008 | 0 | 4 | 4 | 0 | 0 | 0 | 0 | 0 | 0 | 0 | 0 | 0 | 8 |
| 2009 | 2 | 0 | 0 | 0 | 0 | 0 | 0 | 0 | 0 | 0 | 0 | 0 | 2 |
| 2010 | 0 | 0 | 0 | 0 | 0 | 0 | 0 | 0 | 0 | 0 | 8 | 4 | 12 |
| 2011 | 6 | 4 | 12 | 14 | 8 | 4 | 6 | 8 | 4 | 8 | 6 | 2 | 82 |
| 2012 | 0 | 1 | 3 | 0 | 0 | 0 | 56 | 77 | 60 | 44 | 77 | 56 | 374 |
| 2013 | 65 | 5 | 4 | 0 | 0 | 0 | 0 | 5 | 0 | 0 | 4 | 4 | 87 |
| 2014 | 4 | 0 | 0 | 2 | 2 | 10 | 0 | 1 | 0 | 0 | 0 | 38 | 57 |
| 2015 | 86 | 69 | 82 | 71 | 0 | 50 | 42 | 106 | 88 | 145 | 138 | 128 | 1005 |
| 2016 | 116 | 78 | 28 | 71 | 87 | 41 | 144 | 162 | 115 | 180 | 159 | 153 | 1334 |
| 2017 | 77 | 56 | 2 | 9 | 4 | 10 | 13 | 32 | 7 | 3 | 14 | 159 | 386 |
| 2018 | 140 | 150 | 90 | 86 | 74 | 34 | 10 | 11 | 5 | 12 | 0 | 0 | 612 |
| 2019 | 0 | 0 | 0 | 0 | 2 | 9 | 0 | 0 | 0 | 0 | 0 | 0 | 11 |
| 2020 | 0 | 0 | 0 | 0 | 1 | 2 | 12 | 8 | 0 | 0 | 0 | 2 | 25 |
| 2021 | 7 | 1 | 2 | 0 | 0 | 0 | 0 | 0 | 2 | 0 | 2 | 3 | 17 |
| 2022 | 2 | 10 | 0 | 0 | 0 | 0 | 0 | 4 | 4 | 6 | 2 | 0 | 28 |

Figure R3. The number of IAGOS data sampled in the defined region as Southeast Asia.

The IAGOS data use policy is here: https://iagos.aeris-data.fr/data-policy/

We ask you to inform the data providers, traceable through the metadata connected to the provided DOI, when the data is used for publication(s), and to offer them the possibility to comment and/or offer them co-authorship or acknowledgement in the publication when this is justified by the added value of the data for your results. In accordance with the IAGOS data policy, users of IAGOS data products are required to:
1. include the following acknowledgements in publications: "MOZAIC/CARIBIC/IAGOS data were created with support from the European Commission, national agencies in Germany (BMBF), France (MESR), and the UK (NERC), and the IAGOS member institutions (http://www.iagos.org/partners). The participating airlines (Lufthansa, Air France, Austrian, China Airlines, Hawaiian Airlines, Air Canada, Iberia, Eurowings Discover, Cathay Pacific, Air Namibia, Sabena) supported IAGOS by carrying the measurement equipment free of charge since 1994. The data are available at http://www.iagos.fr thanks to additional support from AERIS."
2. offer co-authorship to the IAGOS Principal Investigators if the IAGOS data play a significant role in the publication
3. identify themselves and provide contact information (valid email address)
4. provide a short description of the intended research

Thanks for your notice. We revised our manuscript by acknowledging these data sources.

Beijing ozonesonde trends: The supplement shows decreasing ozone above Beijing, according to the Beijing ozonesonde record (Figures S3-S6). This decreasing trend is the opposite of the positive trends reported by other studies using IAGOS data (Gaudel et al., 2020; Lu et al., 2024). Furthermore, Figure S2 shows a very unusual decrease of ozone after 2011 above Beijing, that I have not seen at any other ozonesonde site. Please download all of the IAGOS ozone profiles above northeastern China (1994- 2022) and compare them to the Beijing ozonesondes to identify the discrepancy.

Thanks for your valuable comments. Actually we were also curious if this negative trend in Beijing is convincing and agree with your statements of "The sharp ozone decrease from the year 2011 is unusual". But we think that we need to discuss about this ozone decrease pattern in Beijing.

1) At first, in the reference papers that you suggested (Gaudel et al., Sci. Adv., 2020; Lu et al., egusphere, 2024), trend analyses using the IAGOS data were performed in the northeast Asia region, not the China or specific city area in China (e.g., Figure 1 in Gaudel et al., 2020 defined the box region as 'Northeast China/Korea, and Figure 1 in Lu et al., 2024 defined the box region as the 'East Asia')

2) We used the IAGOS 'vertical profile' data around the airport. In East Asia, total 11 airports provide this data, as shown in Figure R4. Using this data, we examined the vertical profile of ozone trends for each airport as shown in Figure R5. In Figure R4, you can see that most of airports show the increasing or no trends. Exception cases are Beijing and Qingdao, which are located in the eastern China (Figure R4). In fact, decreasing trends in these two airports have p-value less than 0.05 in the boundary layer (0 to 2 km altitude).

3) We also investigated the time series of ozone at 1, 3, 5, and 7 km altitudes for all 11 airports, and confirmed that ozone in Beijing and Qingdao actually shows the decreasing trend (Figure R6), different from trends at other airports showing increasing trends. Our IAGOS trend analyses in the 'Northeast' and 'Southeast' Asia regions also show clear increasing trends, consistent with previous studies (e.g., Gaudel et al., 2020). But in detail, as we showed, IAGOS data also indicate the 'decreasing trend of ozone' for some regions in eastern China.

Unfortunately, IAGOS does not provide enough data from 2011 to 2013, the period showing sharp ozone decrease based on ozonesonde measurements in Beijing. At this present moment, it is hard to evaluate this sharp ozone drop (from ozonesonde data) during 2011-2013 is meaningful or not. However, comparison with the IAGOS data (as suggested) show that the ozone decrease in Beijing may be a real signal, while the quantity is not clear.

4) Used ozonesonde data at the Beijing site were reported previously in Zhang et al., (2021) (in terms of total column analyses, though). Figure R7 is a figure taken from Zhang et al. (2021), which is the temporal variation of ozone from surface to stratosphere. In their figure, we can see that the tropospheric ozone enhancement during about 2006-2012 looks related to the stratospheric ozone intrusion. This is in line with a number of studies indicating the possible impact of stratospheric ozone intrusion to the tropospheric ozone level in East Asia (e.g., Hua et al., 2024; Koo et al., 2024; Zhao et al., 2024) and a very recent study reported that this stratospheric ozone intrusion becomes weaker in China (declining trend in 2015-2022) (Chen et al., 2024). We do not know that the extent of this stratospheric ozone intrusion can lead the tropospheric ozone decline in China, and this topic is also beyond the scope of this paper. Just

here we would mention that it is hard to neglect the ozone decreasing trend in Beijing that we found based on the ozonesonde and IAGOS data analysis.

In this context, we think that it may be worth to report this ozone decrease in Beijing. But we do not provide features in detail in the manuscript because it seems beyond of the scope of this paper. Now we would leave our discussion here in the response document, and will do some more study in the future. Some statements have been added in Lines 462-467:

"Although trend values are not largely evident, tropospheric ozone decrease at Beijing is quite consistent in all seasons. Zhang et al. (2021) also treated the variation of ozonesonde measurements at Beijing, and it looks that the stratospheric ozone intrusion is strong from 2006 to 2012 but not in other years, which may be related to the ozone trend at Beijing. At this present moment, these decreasing trends were not well explained by our knowledge. Nonetheless, we would report these trends because it can be a motivation of further research."

**References:**

Chen, Z., and co-authors (2024), Stratospheric influence on surface ozone pollution in China. Nature Communications, 15, 4064, https://doi.org/10.1038/s41467-024-48406-x.

Hua, J., and co-authors (2024), Unravelling the impacts of stratospheric intrusions on near-surface ozone during the springtime ozone pollution episodes in Lhasa, China, Atmospheric Research, 311, 107687, https://doi.org/10.1016/j.atmosres.2024.107687.

Koo, J.-H., and co-authors (2024), The Analysis of summertime tropospheric ozone at Anmyeon using ozonesonde measurements in 2021~2022, Journal of Korean Society for Atmospheric Environment, 40(3), 373-383, doi:10.5572/KOSAE.2024.40.3.373

Zhao, K., and co-authors (2024), Impact of stratospheric intrusions on surface ozone enhancement in Hong Kong in the lower troposphere: Implications for ozone control strategy, Atmospheric Environment, 329, 120539, https://doi.org/10.1016/j.atmosenv.2024.120539.

[Figure]

| IATA | City | Latitude | Longitude |
|---|---|---|---|
| PEK | Beijing | 40.39 | 115.83 |
| TAO | Qingdao | 36.34 | 120.91 |
| ICN | Seoul | 35.10 | 126.17 |
| NKG | Nanjing | 32.33 | 117.14 |
| PVG | Shanghai | 30.79 | 122.00 |
| KIX | Osaka | 34.84 | 134.51 |
| MNL | Manila | 15.13 | 120.68 |
| DMK | Bangkok | 14.40 | 99.70 |
| BKK | Bangkok | 14.35 | 100.83 |
| SGN | Ho Chi Minh City | 11.54 | 107.43 |
| SIN | Singapore | 2.02 | 104.20 |

Figure R4. (Left) The location of 11 East Asia airports providing ozone vertical profile data in the IAGOS dataset (https://iagos.aeris-data.fr/download/). (Right) Latitude and Longitude of 11 airports (city name and airport IATA code).

[Figure]

**Figure R5**. Vertical profiles of ozone trends for 11 IAGOS airports in East Asia.

[Figure]

**Figure R6**. Time series of annual median of ozone at 1, 3, 5, and 7 km altitudes for 11 IAGOS airports in East Asia.

[Figure]

**Figure R7**. Temporal variation (left) and vertical mean profile of ozone measured by the ozonesonde experiments at Beijing site. These figures were taken from Zhang et al., (2021) (DOI:10.1088/1748-9326/ac109f).

Table S1: Following the TOAR-II statistics guidelines, all trends need to be reported with the 95% confidence interval and the p-value.

Added. Thanks!

Figure S2 This figure is very difficult to read because it is so small. Please enlarge.

We have improved the quality of this Figure now.

**References**

Chang, K.-L., Cooper, O. R., Gaudel, A., Petropavlovskikh, I., Effertz, P., Morris, G., and McDonald, B. C.: Technical note: Challenges in detecting free tropospheric ozone trends in a sparsely sampled environment, Atmos. Chem. Phys., 24, 6197–6218, https://doi.org/10.5194/acp-24-6197-2024, 2024.

Gaudel, A., Bourgeois, I., Li, M., Chang, K.-L., Ziemke, J., Sauvage, B., Stauffer, R. M., Thompson, A. M., Kollonige, D. E., Smith, N., Hubert, D., Keppens, A., Cuesta, J., Heue, K.-P., Veefkind, P., Aikin, K., Peischl, J., Thompson, C. R., Ryerson, T. B., Frost, G. J., McDonald, B. C., and Cooper, O. R. (2024), Tropical tropospheric ozone distribution and trends from in situ and satellite data, Atmos. Chem. Phys., 24, 9975–10000, https://doi.org/10.5194/acp-24-9975-2024

Lu, X., Liu, Y., Su, J., Weng, X., Ansari, T., Zhang, Y., He, G., Zhu, Y., Wang, H., Zeng, G., Li, J., He, C., Li, S., Amnuaylojaroen, T., Butler, T., Fan, Q., Fan, S., Forster, G. L., Gao, M., Hu, J., Kanaya, Y., Latif, M. T., Lu, K., Nédélec, P., Nowack, P., Sauvage, B., Xu, X., Zhang, L., Li,

K., Koo, J.-H., and Nagashima, T.: Tropospheric ozone trends and attributions over East and Southeast Asia in 1995–2019: An integrated assessment using statistical methods, machine learning models, and multiple chemical transport models, EGUsphere [preprint], https://doi.org/10.5194/egusphere-2024-3702, 2024.

Stauffer, R. M., Thompson, A. M., Kollonige, D. E., Komala, N., Al-Ghazali, H. K., Risdianto, D. Y., Dindang, A., Fairudz bin Jamaluddin, A., Sammathuria, M. K., Zakaria, N. B., Johnson, B. J., and Cullis, P. D.: Dynamical drivers of free-tropospheric ozone increases over equatorial Southeast Asia, Atmos. Chem. Phys., 24, 5221–5234, https://doi.org/10.5194/acp-24-5221-2024, 2024.

We added these references and cited them properly in the manuscript.